# Assessing the vulnerability of marine life to climate change in the Pacific Islands region

**Jonatha Giddens**[1,2‡], **Donald R. Kobayashi**[3‡*], **Gabriella N. M. Mukai**[1,4‡], **Jacob Asher**[1,5☉], **Charles Birkeland**[4☉], **Mark Fitchett**[6☉], **Mark A. Hixon**[4☉], **Melanie Hutchinson**[1☉], **Bruce C. Mundy**[7☉], **Joseph M. O'Malley**[3☉], **Marlowe Sabater**[6☉], **Molly Scott**[1☉], **Jennifer Stahl**[1☉], **Rob Toonen**[8☉], **Michael Trianni**[3☉], **Phoebe A. Woodworth-Jefcoats**[3☉], **Johanna L. K. Wren**[3☉], **Mark Nelson**[9☉]

1 Cooperative Institute for Marine and Atmospheric Research, University of Hawai'i at Mānoa, Honolulu, Hawai'i, United States of America, 2 National Geographic Society Exploration Technology Lab, Washington, DC, United States of America, 3 National Oceanic and Atmospheric Administration, National Marine Fisheries Service, Pacific Islands Fisheries Science Center, Honolulu, Hawai'i, United States of America, 4 School of Life Sciences, University of Hawai'i at Mānoa, Honolulu, Hawai'i, United States of America, 5 The Red Sea Development Company, Riyadh, KSA, 6 Western Pacific Regional Fishery Management Council, Honolulu, Hawai'i, United States of America, 7 Ocean Research Explorations, Honolulu, Hawai'i, United States of America, 8 Hawai'i Institute of Marine Biology, University of Hawai'i at Mānoa, Kāne'ohe, Hawai'i, United States of America, 9 Office of Science and Technology, National Oceanic and Atmospheric Administration, Silver Spring, Maryland, United States of America

☉ These authors contributed equally to this work.
‡ JG, DRK, and GNMM are Joint Senior Authors in this work.
* donald.kobayashi@noaa.gov

**Data Availability Statement:** All relevant data are within the article and its Supporting information files.

## Abstract

Our changing climate poses growing challenges for effective management of marine life, ocean ecosystems, and human communities. Which species are most vulnerable to climate change, and where should management focus efforts to reduce these risks? To address these questions, the National Oceanic and Atmospheric Administration (NOAA) Fisheries Climate Science Strategy called for vulnerability assessments in each of NOAA's ocean regions. The Pacific Islands Vulnerability Assessment (PIVA) project assessed the susceptibility of 83 marine species to the impacts of climate change projected to 2055. In a standard Rapid Vulnerability Assessment framework, this project applied expert knowledge, literature review, and climate projection models to synthesize the best available science towards answering these questions. Here we: (1) provide a relative climate vulnerability ranking across species; (2) identify key attributes and factors that drive vulnerability; and (3) identify critical data gaps in understanding climate change impacts to marine life. The invertebrate group was ranked most vulnerable and pelagic and coastal groups not associated with coral reefs were ranked least vulnerable. Sea surface temperature, ocean acidification, and oxygen concentration were the main exposure drivers of vulnerability. Early Life History Survival and Settlement Requirements was the most data deficient of the sensitivity attributes considered in the assessment. The sensitivity of many coral reef fishes ranged between Low and Moderate, which is likely underestimated given that reef species depend on a biogenic habitat that is extremely threatened by climate change. The standard assessment methodology originally developed in the Northeast US, did not capture the additional complexity of the Pacific region, such as the diversity, varied horizontal and vertical distributions, extent of

**Funding:** The author(s) received no specific funding for this work.

**Competing interests:** The authors have declared that no competing interests exist.

coral reef habitats, the degree of dependence on vulnerable habitat, and wide range of taxa, including data-poor species. Within these limitations, this project identified research needs to sustain marine life in a changing climate.

## Introduction

Climate change is occurring now, affecting our oceans and the species, including ourselves, that depend on marine ecosystems [1–7]. Marine life faces a range of stressors that negatively affect ecosystems and fisheries, including increasing temperature, ocean acidification, decreasing oxygen concentration, and increasing pollution [5, 8, 9]. However, it is not clear which species in particular are most vulnerable to climate change and where science and management should focus efforts to reduce these risks [10].

Until recently, studies of climate change impacts to ocean ecosystems have tended to focus on model species, or single-pressure impacts [11]. To move towards an ecosystem-based approach to management (EBM) we need studies that synthesize knowledge across species [12]. By amassing climate change data across a range of species and environmental variables in a region, we can use the best available information to predict which species are most vulnerable, and why [11]. This information is critical to prioritize ocean ecosystem management and research in a rapidly changing world [13]. The importance of climate change science in fisheries management is highlighted in a recent US National Oceanic and Atmospheric Administration (NOAA) Fisheries guidance document, the "Ecosystem-Based Fisheries Management Road Map" [14]. Ecosystem-based fisheries management, or EBFM, strives to be as inclusive as possible when considering stock dynamics, harvest, and environmental variability, and is an important precursor to EBM [15]. Understanding climate change impacts is a prominent component of the EBFM Road Map.

The Rapid Vulnerability Assessment (RVA) is one method for evaluating multiple impacts to ecosystems to assess climate vulnerability in an EBM context. The RVA is a useful tool for bringing the best available science into a usable format for timely, science-based management by gathering existing data and knowledge through expert opinion and literature searches. This RVA approach identifies data gaps and can inform future research priorities [11].

An RVA methodology was first developed and implemented in the NOAA Fisheries New England/Mid-Atlantic Region to assess the vulnerability of marine life to climate change [16, 17]. In 2015, the NOAA Fisheries Climate Science Strategy called for an RVA to be deployed in each of the remaining NOAA management regions (Southeast, West Coast, Alaska, and Pacific Islands) [18, 19]. The Pacific Islands Vulnerability Assessment (PIVA) project implemented this standard RVA method using expert knowledge, literature review, and climate models to assess the relative vulnerability of 83 marine taxa within the Pacific Islands Region following the methodology of Hare et al. [16].

In accordance with the RVAs implemented across NOAA management regions, PIVA focused on a short time horizon (to 2055). The PIVA included a wide range of taxa, from highly migratory fish species to site-attached invertebrates. Within this range of adult mobility, many of the coral reef species assessed depend upon a biogenic habitat that is itself extremely threatened by climate change.

The Pacific region offered unique challenges including the wide range of taxa, many data-poor species, multiple ecosystems, the diversity and extent of coral reef habitats, the large spatial domain considered, and dependence on vulnerable habitat. As such, the Pacific region

**Table 1. Numbers of species per functional group assessed in the Pacific Islands Vulnerability Assessment (PIVA).**

| Functional Group | Number of Species |
|---|---|
| Pelagic | 6 |
| Shark | 6 |
| Deep Slope | 11 |
| Coastal | 6 |
| Coral Reef | 42 |
| Invertebrate | 12 |

posed additional complexity that was not captured in the standard assessment methodology originally developed in the Northeast United States. Within the RVA limitations, here we provide a relative climate vulnerability ranking across species, identify key attributes and factors that drive this vulnerability, and identify critical data gaps in understanding and mitigating climate change impacts to marine life in the Pacific.

## Methods

### Taxonomic scope

This assessment focused on 83 species from six functional groups (Table 1 and S1 Table) and 33 families across a wide range of locations in the Central, West, and South Pacific Ocean (Hawaiian Archipelago, American Samoa, Mariana Islands Archipelago, including Guam and the Commonwealth of the Northern Mariana Islands, and the Pacific Remote Island Areas [PRIAs] of Baker Island, Howland Island, Jarvis Island, Johnston Atoll, Kingman Reef, Palmyra Atoll, and Wake Atoll) (Fig 1). While each species was assessed individually, results were displayed by functional groups generally based on range size and habitat association as follows: pelagic, shark, deep-slope, coastal, coral reef, and invertebrate species. Because the "coral reef" group of fishes contained the most species, this category was further divided into JEGS (Jacks, Emperors, Groupers, Snappers), Parrotfishes, Surgeonfishes, and "other coral reef" fishes (S2 Table).

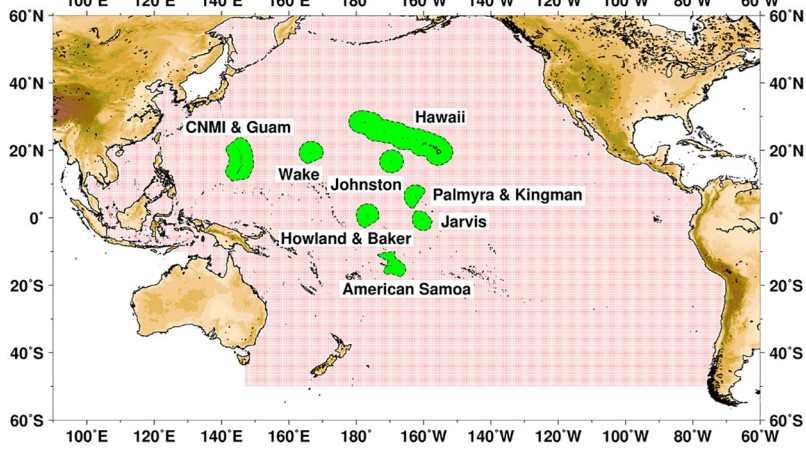

**Fig 1. Map of the Pacific region, and areas included in PIVA.** Green shading denotes extent of the Exclusive Economic Zone around each location. Stippled area denotes the Pacific spatial domain of environmental variables considered when sub-setting for specific taxa ranges.

The species included in the assessment were selected based on regional expert opinion, stock status, commercial and recreational catch records of food fish, and cultural and conservation importance. The final selection also took into account key species representing important ecosystem functions. Throughout this assessment we followed Fishbase.org taxonomy listings and the American Fisheries Society style guide for capitalization of species common names in the text [20]. Ecological research on the Brown Chub *Kyphosus bigibbus* available for this RVA followed an earlier classification in which the species was considered to be a widespread Indo-Pacific species, so we did not incorporate the findings of the latest taxonomic revision of the genus [21].

Climate change vulnerability, as defined here, is composed of both exposure variables and sensitivity attributes. Exposure and sensitivity are defined as separate components of vulnerability below:

## Exposure

The term "exposure," as used in our PIVA, measures the degree to which a species is likely to experience a detrimental change in a particular physical variable in its environment as a consequence of climate change. These variables were selected based on local expert knowledge as the most biologically relevant factors that would affect species in the Pacific Islands Region. Variables were also constrained by the requirement that future projections were available for the time scale of interest. The final list of variables considered in PIVA were temperature (surface, bottom), salinity (surface, bottom), ocean acidification (pH), mixed layer depth, precipitation, current velocity, wind stress, surface oxygen, sea level rise, wave height, chlorophyll, and primary productivity. Exposure variables were tabulated into a binary relevance matrix for each species. The majority of these exposure variables were available in a global gridded format, facilitating the spatial matching of exposure variables to the known geographical distributions of particular species. Exposure variables that were not available in global gridded format were examined using average values, percent changes, or projected ranges of change due to climate impacts.

To characterize the expected climate change in a particular region of the ocean in a method consistent with established RVA protocol [16], grids of standardized anomalies for all exposure variables were downloaded and subsetted to match the current distribution of each species. Species distribution maps were obtained from online sources (OBIS [22]; fishbase.org [23]; sealifebase.org [24]) and gridded for use as masks to subset the exposure variable grids. Global grids of modeled exposure variable standardized anomalies were obtained from the NOAA Earth System Research Laboratory Climate Change Web Portal at https://www.esrl.noaa.gov/psd/ipcc/ocn/ [25]. Averages of all available model runs (n = 23) were used for each variable under consideration using the climate change forecast scenario of RCP8.5 representing the highest greenhouse gas emission scenario [26]. Model runs in this context refers to all independent data outputs available from CMIP5 [25]. Data were represented in 0.5-degree latitude and longitude resolution grids. Standardized anomalies of future climate change were calculated relative to historical values using the data grids. This standardized anomaly represents the difference in the mean climate in the future time period (2006–2055) compared to the historical reference period (1956–2005), standardized by the de-trended interannual standard deviation for the historical reference period (1956–2005). Both the future and historical periods were generated from the same climate models. In short, this represents the projected amount of change in units of historical standard deviation. This particular standard deviation is reflective of the multi-decadal variability of any particular exposure variable at a given point in space (0.5-degree resolution) but excludes the variability associated with any interannual

trend (hence the standardization step using the de-trended interannual standard deviation). The latter point is critical because the standardized anomaly is intended to represent the nature of a large-scale trend across two long, albeit adjacent, blocks of time. As an additional step done for the PIVA, the standardized anomaly was further examined with respect to 1) the amount of change projected to occur and 2) the historical pattern of variability in the 1956–2005 reference period, because these two components enter into the standardized anomaly calculation. This specific exercise was useful to better understand how the standardized anomaly value can be driven both by patterns of absolute change in an exposure variable and by patterns of historical variability, or notably lack thereof of the latter. Locally downscaled data products were not pursued for this exercise based on the following: most of the taxa considered have large geographic ranges covering many degrees of latitude and longitude, and; we wanted to use consistent data products (i.e., ESRL climate data portal https://www.esrl.noaa.gov/psd/ipcc/ocn/ [25]) across all of the NOAA National Marine Fisheries Service climate RVA projects (both already undertaken as well as those currently ongoing) for purposes of regional comparisons.

Pacific-wide exposure was considered in our final analysis consistent with RVA protocol; however, spatial subsets of each exposure variable and taxon were created to better characterize the regional variability in exposure for the wide-ranging PIVA taxa. Four such subsets were created to examine exposure in the Hawaiian Archipelago, American Samoa, Mariana Islands Archipelago (including Guam and the Commonwealth of the Northern Mariana Islands), and a portion of the Pacific Basin that encompassed the U.S. Pacific Remote Island Areas (PRIAs) of Baker Island, Howland Island, Jarvis Island, Johnston Atoll, Kingman Reef, Palmyra Atoll, and Wake Island. The spatial domain for each of these subsets, as well as the larger Pacific-wide domain, examined are presented in Table 2. Note that exposure variable scoring tabulations did not take place over these entire spatial domains, but rather only where the spatial domain intersected the known distribution for a particular taxon, using a 200 nautical mile exclusive economic zone (EEZ) as an additional mask for the regional subsets.

Individual summary visualizations were created for each of the spatial domain, taxon, and exposure variable combinations. For each combination, exposure scores (Low = 1, Moderate = 2, High = 3, Very High = 4) were assigned to each pixel in the spatial domain and tabulated (see S1 File for cutoffs). Each summary included a pair of histograms for these exposure tabulations with respect to signage of projected perturbations relative to baseline (negative for decreases, positive for increases) and overall exposure combining positive and negative under the assumption that both types of perturbations could be assessed together (e.g., a large decrease or a large increase could be equally deleterious). Positive and negative perturbations were aggregated and their exposure scores were averaged and included on the visualization (S1 File).

**Table 2. Specification of spatial domains utilized in PIVA.**

| Spatial Domain | Eastern Extent | Western Extent | Southern Extent | Northern Extent |
|---|---|---|---|---|
| Pacific | 90˚E | 60˚W | 60˚S | 60˚N |
| Samoa Archipelago | 180˚E | 160˚W | 20˚S | 8˚S |
| Hawaiian Archipelago | 172˚E | 145˚W | 11˚N | 36˚N |
| Mariana Archipelago | 132˚E | 158˚E | 10˚N | 25˚N |
| Pacific Remote Island Areas | 154˚E | 152˚W | 5˚S | 27˚N |

These boundaries were then subset by geographic ranges of a specific taxon, and for regional applications were further delineated by USA EEZs.

Exposure variables that were not available with globally gridded data products (e.g., wave height, sea level rise) were addressed in narrative form using the best available scientific information (e.g., https://cmgwindwave.usgsportals.net for modeled wave height data [27], and https://podaac.jpl.nasa.gov/dataset/RECON_SEA_LEVEL_OST_L4_V1 for sea level height [28]). The gridding, subsetting, and tabulating of exposure variables and the resulting visualizations were accomplished with scripting code in the free software package Generic Mapping Tools [29]. The scripting code is freely available from the authors upon request.

## Sensitivity

Sensitivity is a biological trait-based variable, which, for this study, was informed by extensive literature review and expert opinion. Twelve sensitivity attributes, consistent within Rapid Vulnerability Assessments (RVAs) across all NOAA regions were considered: Habitat Specificity; Prey Specificity; Complexity in Reproductive Strategy; Sensitivity to Ocean Acidification; Early Life History Survival and Settlement Requirements; Dispersal of Early Life Stages; Sensitivity to Temperature; Population Growth Rate; Stock Size/Status; Adult Mobility; Spawning Cycle; and 'Other Stressors' (including habitat degradation, pollution, disease, or changes in the food web).

Expert opinions (n = 15) were solicited and compiled prior to them being refined during a 3-day facilitated workshop in March 2018. Expert scorers were aided by compiled species profiles [30–35], which summarized existing literature on the sensitivity attributes for each species. We note with emphasis that when information was insufficient to make assignments with reasonable confidence, experts used their own discretion when scoring. Five experts assessed each species to obtain replication in biological sensitivity scores. Experts scored attributes using a five-tally scoring system where an expert had five tallies to assign to bins (Low = 1, Moderate = 2, High = 3, Very High = 4). Tallies could be assigned to one bin or be spread across multiple bins indicating uncertainty. Attributes were scored prior to the workshop and experts were allowed to change their responses based on the discussion at the workshop. Scores were then averaged.

To assess a species' propensity for distributional shifts with climate change, a subset of four attributes were evaluated: Adult Mobility, Dispersal of Early Life Stages, Habitat Specificity, and Sensitivity to Temperature.

Experts were asked to identify the six most important sensitivity attributes for each species. Considering the wide variety of taxa included in the assessment, and the unique geography compared to other NOAA regions, some sensitivity attributes had very little information available. To account for this, the "top six" analysis was used to compare sensitivity scores against the full suite of 12 attributes with the aim of reducing uncertainty and better assessing the relative vulnerability rankings of species within the Pacific region. The "top six" analysis highlighted the most important sensitivity attributes. Then the sensitivity attributes were compared to their data quality score (ranked between 0 as no data, and 3 as data-rich) to determine the crucial data gaps in understanding climate threats to Pacific Island region species. The data quality score described the literature available for each sensitivity attribute. A data quality score of 3 means data were observed, modeled or empirically measured for the stock in question and came from a reputable source. A score of 2 means data were from related or similar stocks or species, came from outside the study area, or the reliability of the source was limited. A score of 1 reflects the expert judgment of the reviewer and is based on their general knowledge of the stock, or other related stocks, and their relative role in the ecosystem. A score of 0 means very little was known about the stock or related stocks and there was no basis for forming an expert opinion.

Table 3. Sensitivity and exposure component scores, numeric value, and logic rule thresholds.

| Exposure or Sensitivity Component Score | Numeric Value | Logic Rule Thresholds |
|:---:|:---:|:---:|
| Very High | 4 | 3 or more attributes of factors with mean $\geq 3.5$ |
| High | 3 | 2 or more attributes of factors with $3.0 \leq$ mean $<3.5$ |
| Moderate | 2 | 2 or more attributes of factors with $2.5 \leq$ mean $<3.0$ |
| Low | 1 | All other scores $<2.5$ |

## Vulnerability matrices

The process for calculating an overall vulnerability rank is the same as described in detail in Morrison et al. [17] and Hare et al. [16]. First, a component score was calculated for both exposure and sensitivity given the number of factors/attributes above a certain threshold (Table 3). Then, the overall vulnerability rank was determined by multiplying the exposure and sensitivity component scores. The possible range of these scores is between 1 and 16. The numerical values for the climate vulnerability rank were the following: 1–3 Low, 4–6 Moderate, 8–9 High, and 12–16 Very High (S1 Fig). Overall climate vulnerability was illustrated in color-coded matrices with exposure and sensitivity axes.

A bootstrap analysis was performed in the statistical programming language R [36] to quantify uncertainty in the sensitivity rankings. The bootstrapping process consisted of resampling the tallies with replacement for each exposure factor and sensitivity attribute, then recalculating the component scores and vulnerability ranking for each species. This was conducted 1,000 times and the results across each iteration were sorted into their respective overall vulnerability ranking (i.e., Low, Moderate, High, Very High). The bootstrap analysis demonstrated the uncertainty in point estimates (average expert scores) of the vulnerability rankings and illuminated issues that arose from using simple thresholds in the scoring process.

## Results

### Climate exposure

All species ranked Very High in the overall exposure component of vulnerability when all exposure variables were considered together. These high rankings were driven by three dominant factors: (1) decrease in oxygen concentration [37, 38]; (2) rise in sea surface temperature [39]; and (3) increase in ocean acidification (decrease in surface pH) [40] (Fig 2). Exposure score values were considered for both the projected absolute change in a particular variable as well as how the historical variability can influence the standardized anomaly. For example, with exposure variables that exhibit very little historical variability a very slight offset in absolute value projected for the future can translate to a relatively large standardized anomaly when dividing by a very small historical standard deviation. The high exposure scores for ocean acidification (surface pH; a decrease in pH of 0.1) and surface oxygen concentration (a decrease of 5.2 mmol m-3) should be noted with the caveat that the projected changes for the future are relatively slight for all taxa considered.

Exposure scores also relate closely to the spatial domain examined over the distributional range of a particular taxa. For example, surface oxygen exposure summaries for Bigeye Tuna *Thunnus obesus* show a range from Moderate to Very High negative scores for this variable (Fig 3), which can then be further explored by examination of the four subset domains of US EEZs in the Pacific (Figs 4–7). In this example we can see that the overall surface oxygen exposure score for Bigeye Tuna in the Pacific falls between High and Very High when considering the entire geographic range of this species across the Pacific study region. Most areas within

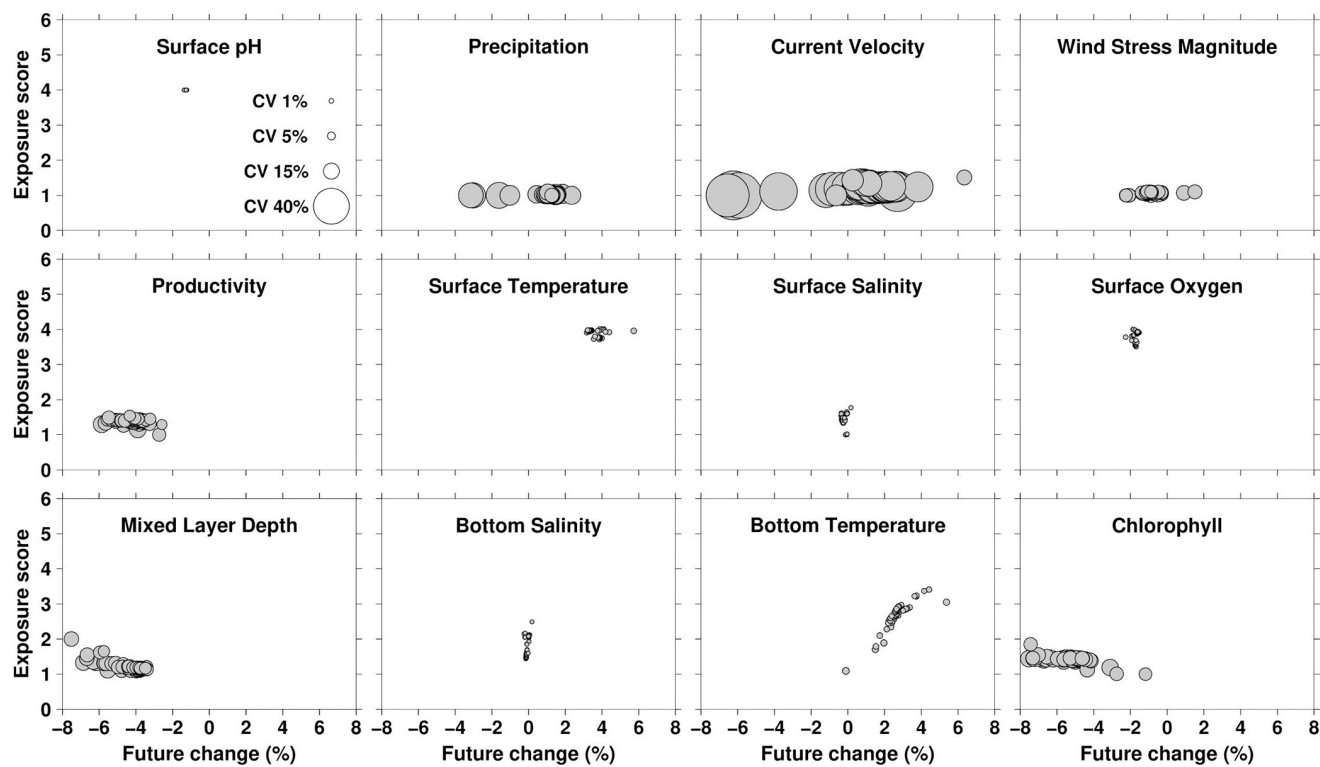

**Fig 2. Exposure scores for 12 variables as related to change over time and historical variability.** Exposure scores for 12 variables plotted on y-axis, change over time plotted on x-axis, and historical variability (coefficient of variation, CV) proportional to the size of symbol. All 83 taxa considered in PIVA are shown in each panel. Symbols are plotted in descending order of variability to minimize symbol overlayment.

the distributional range of this species are projected to experience lower surface oxygen concentrations in the future, but the impact varies spatially. All US EEZs experience High or Very High surface oxygen exposure (Figs 4–7). Regions of moderate exposure are primarily in the high seas away from US EEZs (Fig 3).

In some situations, the scoring summary for an exposure variable can include both positive and negative scores indicating that, across that spatial domain, perturbations in that variable can be either increasing or decreasing relative to baseline reference levels depending on the exact geographic location. In these situations, the absolute values of the bidirectional effects are combined in the upper right panels of the exposure visualizations, with the star on the x-axis representing the average score on a Low, Moderate, High, Very High scale combining both positive and negative scores. An example of this is seen in Fig 8 for chlorophyll exposure and Bigeye Tuna across all habitats in the Pacific domain. An important point is that the positive and negative scores do not "average out" but instead combine together in magnitude to characterize an assumed deleterious effect. The combined and overall exposure effect falls between Low and Moderate for the example of Bigeye Tuna and chlorophyll exposure.

Two exposure variables were not available as a gridded data product under climate projection scenarios: sea-level and wave height. For sea-level rise, estimates were taken from the literature for inclusion in the assessment [41–43]. The latest US Climate Science Special Report [44] indicates that sea level rise will occur within a range of 0.3 m to 2.5 m above a 1991–2009 reference period by the year 2100. While it is difficult to ascertain impacts from this particular exposure variable it could clearly have strong effects on shallow ecosystems, particularly for

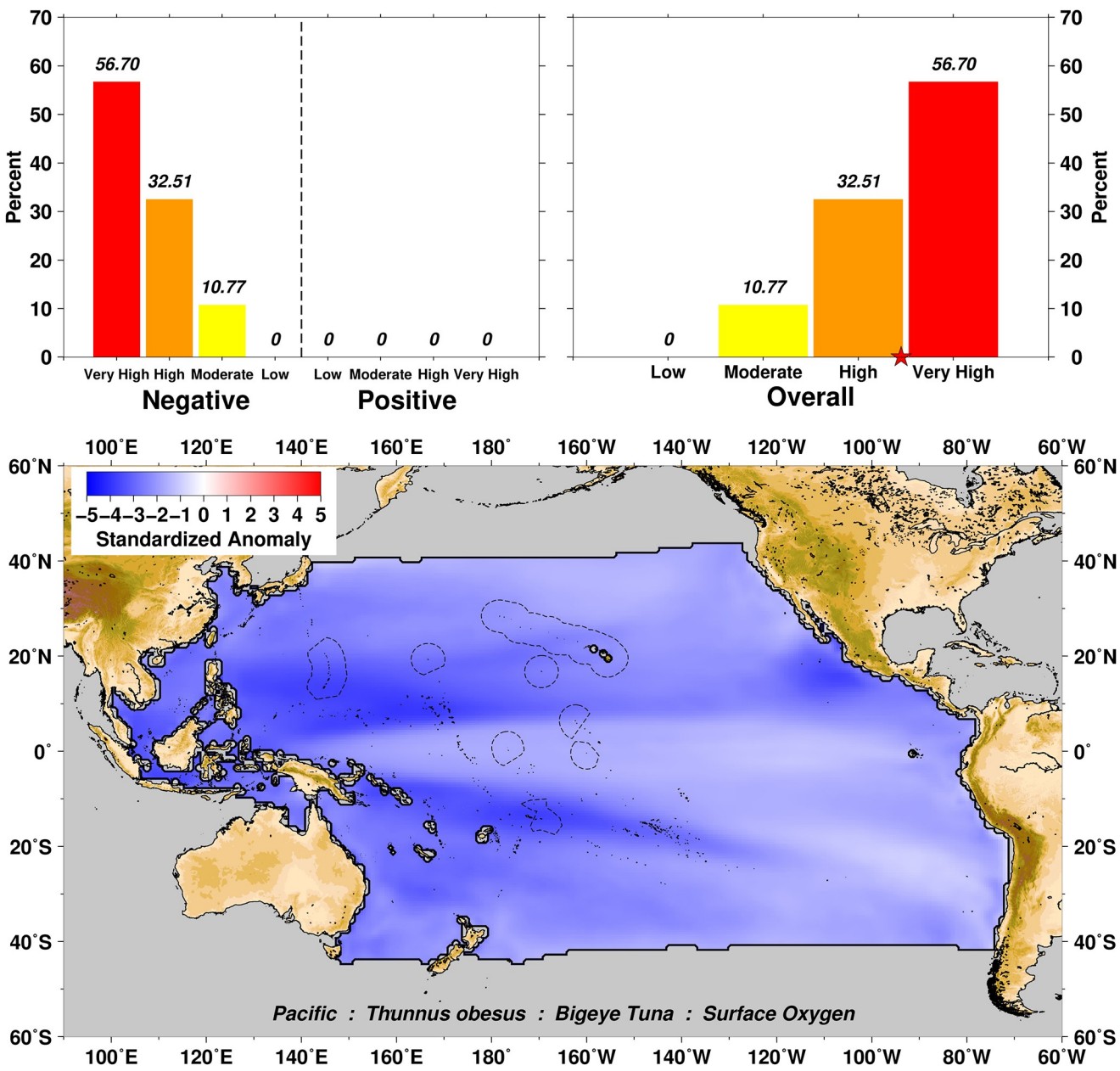

**Fig 3. Surface oxygen exposure summaries for Bigeye Tuna over the Pacific region.** The percentage of habitat experiencing different levels of change are depicted in the upper left histogram (Low = green, Moderate = yellow, High = orange, and Very High = red). The absolute value of changes in positive and negative change are combined in the upper right histogram panel, with the star on the x-axis representing the average exposure score. The bottom panel is the geographic visualization of perturbations to the habitat relative to the baseline reference level (color shaded values of standardized anomaly).

intertidal taxa such as the Hawaiian yellow-foot limpet *Cellana sandwicensis*. This exposure variable needs further scrutiny, especially as it relates to the extent of potential habitat at higher elevations on particular coastlines.

For all five locations there were projected small decreases in mean wave height ranging from -0.06% to -1.81% using the USGS climate change wave projections [45]. Climate change impacts to wave activity vary considerably across many published studies emphasizing both the general uncertainty of wave responses to climate change and their predominantly smaller

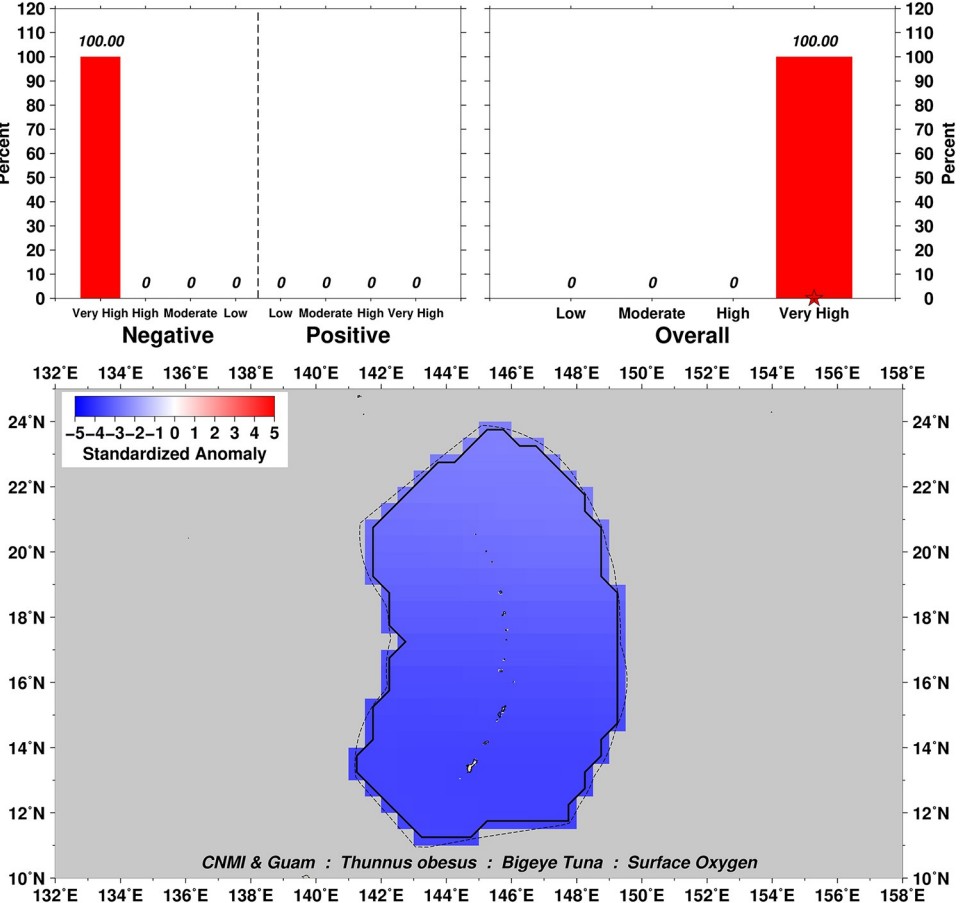

**Fig 4. Surface oxygen exposure scores for Bigeye Tuna in the Commonwealth of the Northern Mariana Islands (CNMI) and Guam located in the Mariana Archipelago.** Colors and histograms interpreted as in Fig 3.

scales of variability over both time and space (e.g., [46, 47]). While it is difficult to determine the overall impact to the different taxa in different regions, these wave height changes would appear to be small relative to most of the other exposure scores examined and would likely only affect the species which inhabit shallow areas. Wave height can vary substantially over a relatively small spatial scale depending on the direction, degree of protection, and seafloor contours [48]. This variable is worthy of further study, as well as including other metrics such as wave direction, rogue wave frequency, and related variables pertaining to storm activity.

## Vulnerability and biological sensitivity

The means of biological sensitivity point estimates ranged between Low and Very High [30–35]. Functional groups that encompassed larger-bodied species generally shared similar sensitivity scores, whereas the groups with smaller and site-attached species were more variable. Results of biological sensitivity and overall vulnerability scores are discussed below by functional group, from the large-bodied and wide-ranging pelagic fishes, to small-bodied low mobility invertebrate species.

## Overall vulnerability

The climate change vulnerabilities for 83 taxa in the Pacific showed a broad pattern across functional groups (S2 File). The larger-bodied and more wide-ranging pelagic and coastal

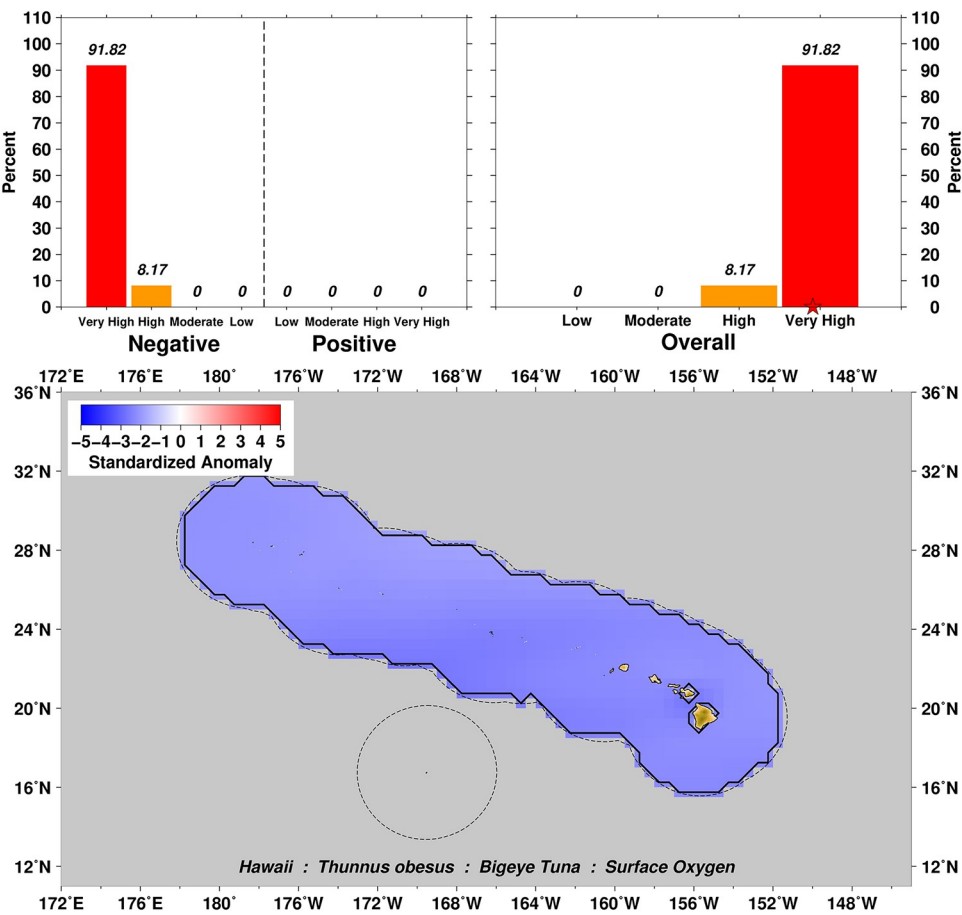

**Fig 5. Surface oxygen exposure scores for Bigeye Tuna in the Hawaiian Archipelago.** Colors and histograms interpreted as in Fig 3.

species not associated with coral reefs ranked lowest in vulnerability, while the smaller-bodied and more site-attached species (small coral reef fishes and invertebrates) ranked highest in vulnerability (Fig 9). Within-group differences were evident as well. For invertebrates, taxa were more evenly distributed between Moderate to Very High vulnerability rankings, whereas for the coastal group, all species fell within the Moderate category.

## Pelagic fishes

Vulnerability ranged from Moderate to High for pelagic teleost species (Fig 10). The Striped Marlin *Kajikia audax* ranked highest in sensitivity because of its low Stock Size/Status score [30].

## Sharks

For sharks, vulnerability ranged from Moderate to Very High. The Oceanic Whitetip Shark *Carcharhinus longimanus*, Scalloped Hammerhead Shark *Sphyrna lewini*, Pelagic Thresher Shark *Alopias pelagicus*, and Silky Shark *Carcharhinus falciformis* were ranked higher in overall sensitivity because both Population Growth Rate and Stock Size/Status were of concern [31] (Fig 11).

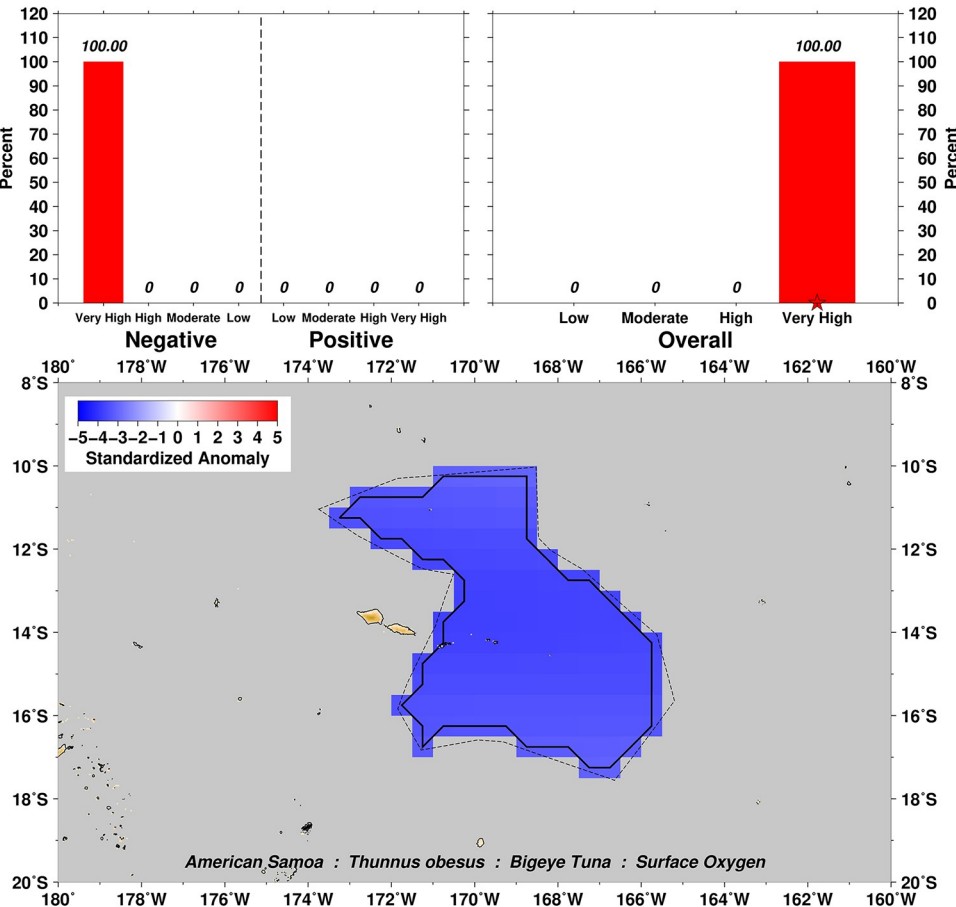

**Fig 6. Surface oxygen exposure scores for Bigeye Tuna in the American Samoa Archipelago.** Colors and histograms interpreted as in Fig 3.

## Deep-slope fishes

Vulnerability of deep-slope species ranged from Moderate to High, with the Deep-Water Red Snapper *Etelis carbunculus*, Hawaiian Grouper *Hyporthodus quernus*, and Slender Armorhead *Pentaceros wheeleri* being most sensitive. These higher scores were due in part to temperature sensitivity and low population growth rate for Hawaiian Grouper, and concern of small stock size for the Slender Armorhead [32] (Fig 12).

## Coastal fishes

Coastal species not associated with coral reefs scored as Moderate in vulnerability. All coastal species ranked Low in overall sensitivity. This group was made up of only six species, and their component sensitivities all ranged from Low to Moderate, with only Bonefish *Albula glosso-donta*, Threadfin *Polydactylus sexfilis*, and Greater Amberjack *Seriola dumerilii* sensitivity attributes falling in the Very High category for any component score [33] (Fig 13).

## Coral reef Jacks, Emperors, Groupers, and Snappers (JEGS)

Vulnerability of coral reef JEGS ranged from Moderate to High, with most species falling in the Moderate category. The Black-tip Grouper *Epinephelus fasciatus* and the Two-spot

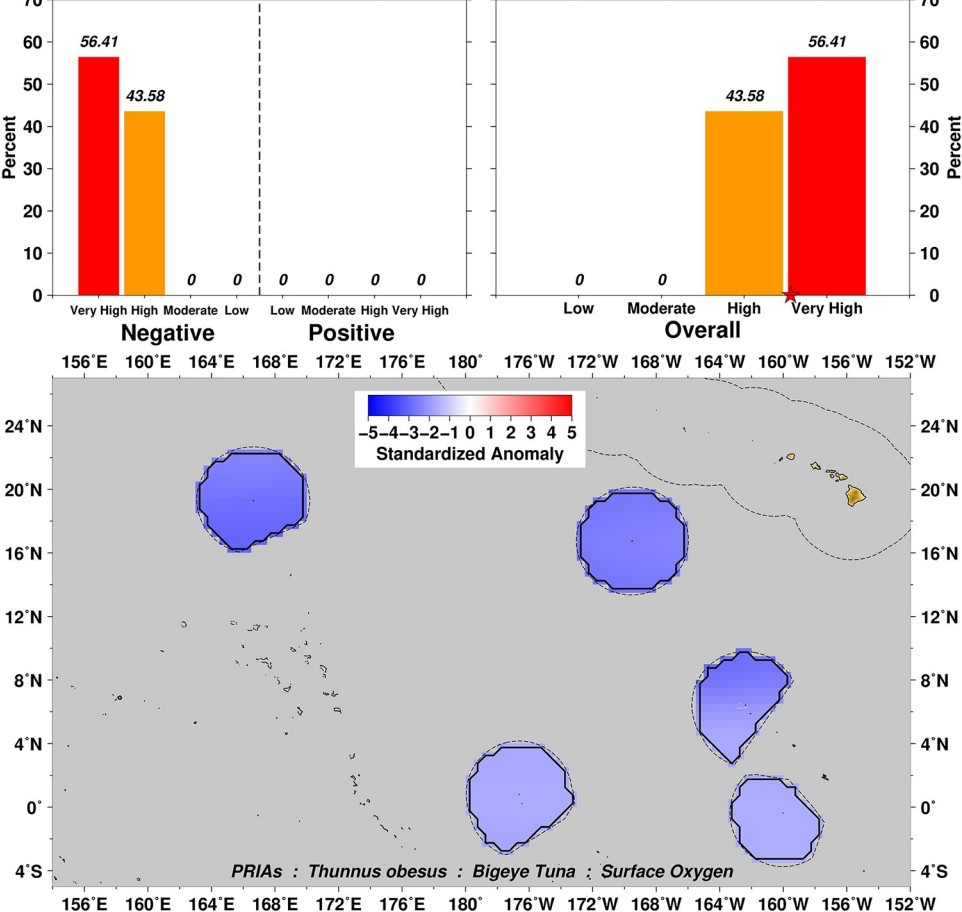

**Fig 7. Surface oxygen exposure scores for Bigeye Tuna in the U.S. Pacific Remote Island Areas (PRIAs).** Colors and histograms interpreted as in Fig 3.

Snapper *Lutjanus bohar* stood out as more sensitive because of complexity in reproduction and specialized early life history requirements [34] (Fig 14).

## Coral reef parrotfishes

Vulnerability ranged from Moderate to High for coral reef Parrotfishes. Only two species ranked High, Bumphead Parrotfish *Bolbometopon muricatum* and Steephead Parrotfish *Chlorurus microrhinos*. The Bumphead Parrotfish stood out with greater sensitivity because of dietary requirements for live coral, habitat specificity, early life history requirements, sensitivity to ocean acidification, and low stock size. The Steephead Parrotfish ranked High in vulnerability because of sensitivity attributes: Habitat Specificity, Early Life History Survival and Settlement Requirements, and Complexity in Reproductive Strategy [34] (Fig 15).

## Coral reef surgeonfishes

Coral reef Surgeonfishes' vulnerability ranged from Moderate (two species) to High (six species) [34] (Fig 16). Stock Size/Status was the most frequent attribute ranked higher in sensitivity for the species with greater vulnerability scores. In addition, the Bluespine Unicornfish

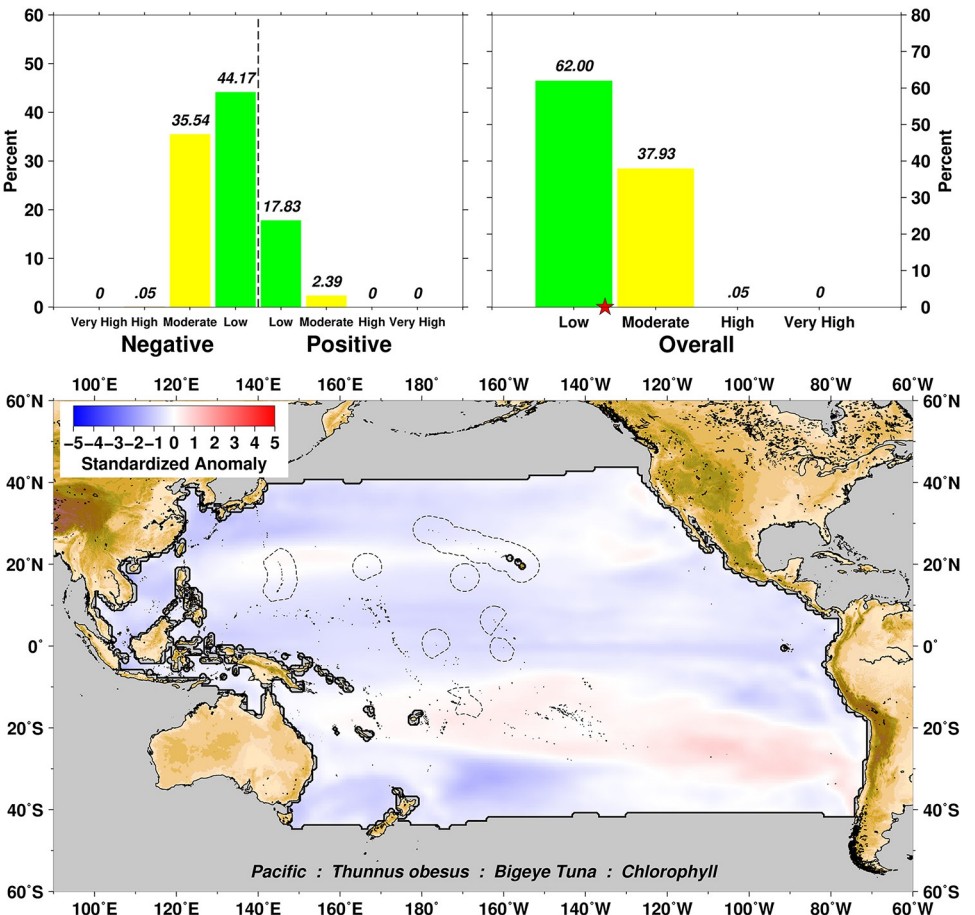

**Fig 8. Chlorophyll exposure scores for Bigeye Tuna over the Pacific region, showing an example of bidirectional exposure scoring.** Colors and histograms interpreted as in Fig 3.

*Naso unicornis* and the Sleek Unicornfish *Naso hexacanthus* had increased sensitivity scores due to low Population Growth Rate.

## Other coral reef fishes

For other coral reef fishes, vulnerability ranged from Moderate to Very High. The Arceye Hawkfish *Paracirrhites arcatus* and the Ornate Butterflyfish *Chaetodon ornatissimus* scored the highest in vulnerability. For Arceye Hawkfish, coral Habitat Specificity and Adult Mobility were the two most sensitive attributes. For the Ornate Butterflyfish, prey coral sensitivity and sensitivity of prey coral to ocean acidification stood out as being of higher concern [34] (Fig 17).

## Invertebrates

The group that scored the highest in vulnerability was the invertebrates. Six species ranked Very High in vulnerability, and one species, the endemic Hawaiian yellow-foot limpet, ranked Very High in sensitivity. For the limpet, this was due to a very high sensitivity score in every attribute except population growth rate [35] (Fig 18).

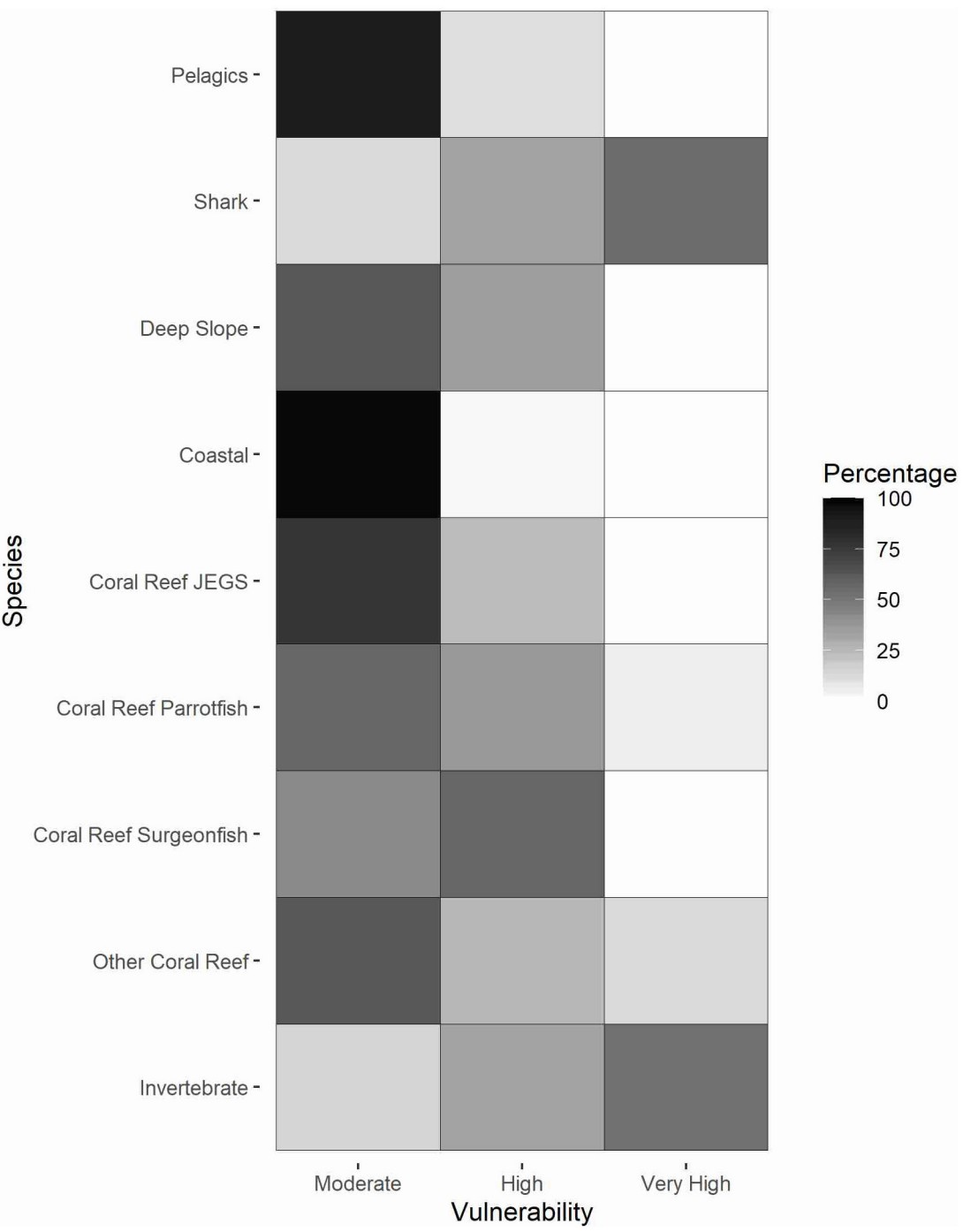

**Fig 9. Summary of vulnerability scores across functional groups.** Vulnerability scores were standardized to represent the percentage of species within the group that fell in a particular category (from Moderate to Very High).

## Bootstrap sensitivity scores and uncertainty

There were fourteen species whose bootstrapped sensitivity scores had low certainty (S3 File). In these instances, the low certainty was defined as less than 66% probability of being categorized as a particular level of vulnerability [16]. These species were Two-spot Snapper (53%

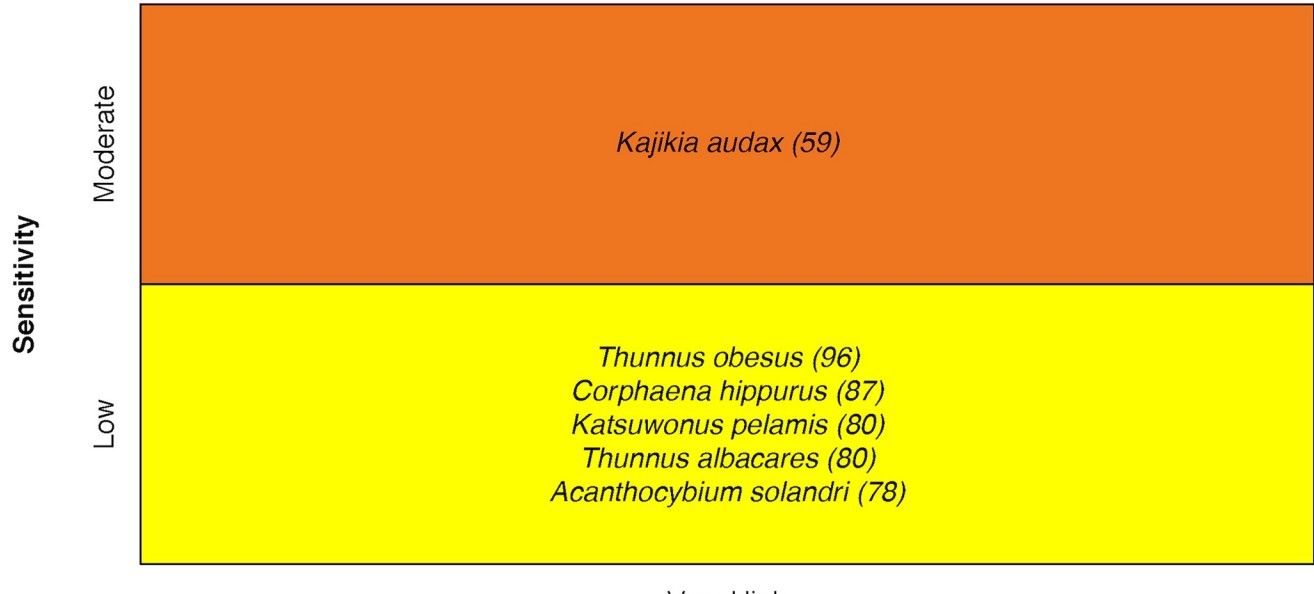

**Fig 10. Vulnerability scores of pelagic fishes.** Overall climate vulnerability, as determined by point estimates, is denoted by box background color: Moderate (yellow), and High (orange). Certainty in vulnerability score is denoted by parenthetical number following the species name, indicating bootstrapped scores (0–100% certainty).

High and 47% Moderate); Green Jobfish Aprion virescens (36% High and 64% Moderate); Bullethead Parrotfish (54% High and 46% Moderate); Bumphead Parrotfish (46% Very High and 54% High); Tanned-faced Parrotfish Chlorurus frontalis (62% High and 38% Moderate); Red-lipped Parrotfish Scarus rubroviolaceus (39% High and 61% Moderate); Spectacled Parrotfish Chlorurus perspicillatus (39% High and 61% Moderate); Achilles Tang *Acanthurus achilles* (51% High and 49% Moderate); Epaulette Surgeonfish *Acanthurus nigricauda* (57% High and 43% Moderate); Sleek Unicornfish (58% High and 42% Moderate); Sabre Squirrelfish *Sargocentron spiniferum* (47% High and 53% Moderate); Striped Marlin (59% High and 38% Moderate; 3% Low); Pelagic Thresher Shark (58% Very High and 42% High); and palolo worm Palola viridis (63% Very High and 37% High). In addition, eight species were placed in a different vulnerability category by the bootstrapped sensitivity analysis than they were originally. The Pink Snapper, Sleek Unicornfish, Epaulette Surgeonfish, Tanned-faced Parrotfish, and Bullethead Parrotfish moved from Moderate to High. The palolo worm moved from High to Very High. The Little Spine Foot moved from High to Moderate. Overall, 47 species' scores were highly certain (greater than or equal to 90%) and 22 species' scores were moderately certain (66–89%).

## Distributional vulnerability rank

Most species ranked Low or Moderate in propensity to shift geographic distributions with climate change. The exceptions were most shark and pelagic species, given their large range of adult mobility. Species that ranked very high were both coastal scads as well as the Giant Trevally *Caranx ignobilis* [30–35].

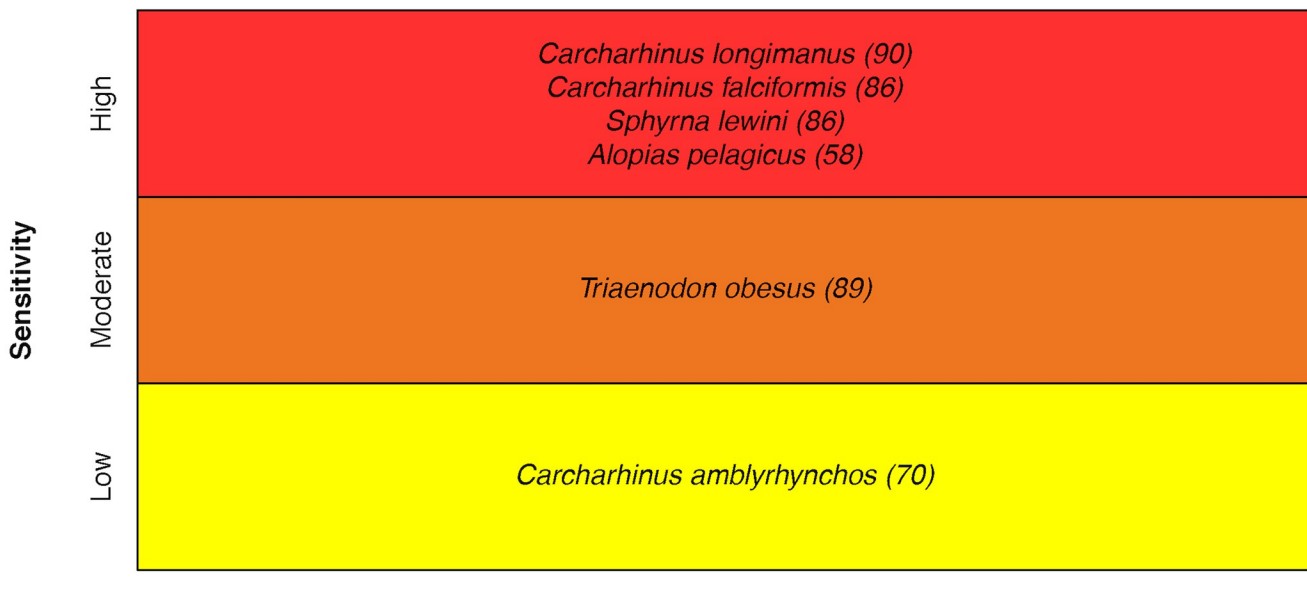

**Fig 11. Vulnerability scores of sharks.** Overall climate vulnerability, as determined by point estimates, is denoted by box background color: Moderate (yellow), High (orange), and Very High (red). Certainty in vulnerability score is denoted by parenthetical number following the species name, indicating bootstrapped scores (0–100% certainty).

### Data gaps

Reducing the number of sensitivity attributes considered for Pacific Island region species from 12 to 6 did not change the overall sensitivity scores. Rather, the scores were driven by the high-ranking sensitivities, and these attributes were consistently chosen as the most important. Therefore, species scores remained similar or higher with all 12 attributes considered. The focused analysis did highlight the important data gaps that require further research to better understand how climate change affects Pacific Island region species. Five attributes emerged as important, yet data deficient. In decreasing order of importance, the percent of experts selecting each was Early Life History Survival and Settlement Requirements (63%), Stock Size/Status (12%), Other Stressors (12%), Complexity in Reproductive Strategy (5%), Population Growth Rate (4%), and Spawning Cycle (1%).

A sensitivity attribute of particular importance in assessing sensitivity to climate was Early Life History Survival and Settlement Requirements. For more than half of the species, all experts were in complete agreement on the importance of this attribute.

## Discussion

### Drivers of vulnerability

**Pelagic and coastal fishes.** There are a variety of important environmental considerations in the pelagic environment with respect to climate change. For example, rising ocean temperatures may be a threat to pelagic species [49, 50]. Skipjack Tuna *Katsuwonus pelamis* may already inhabit the high extreme of their temperature threshold [51], leaving little habitat buffer for expansion. The vertical expansion of the oxygen minimum zone with climate change

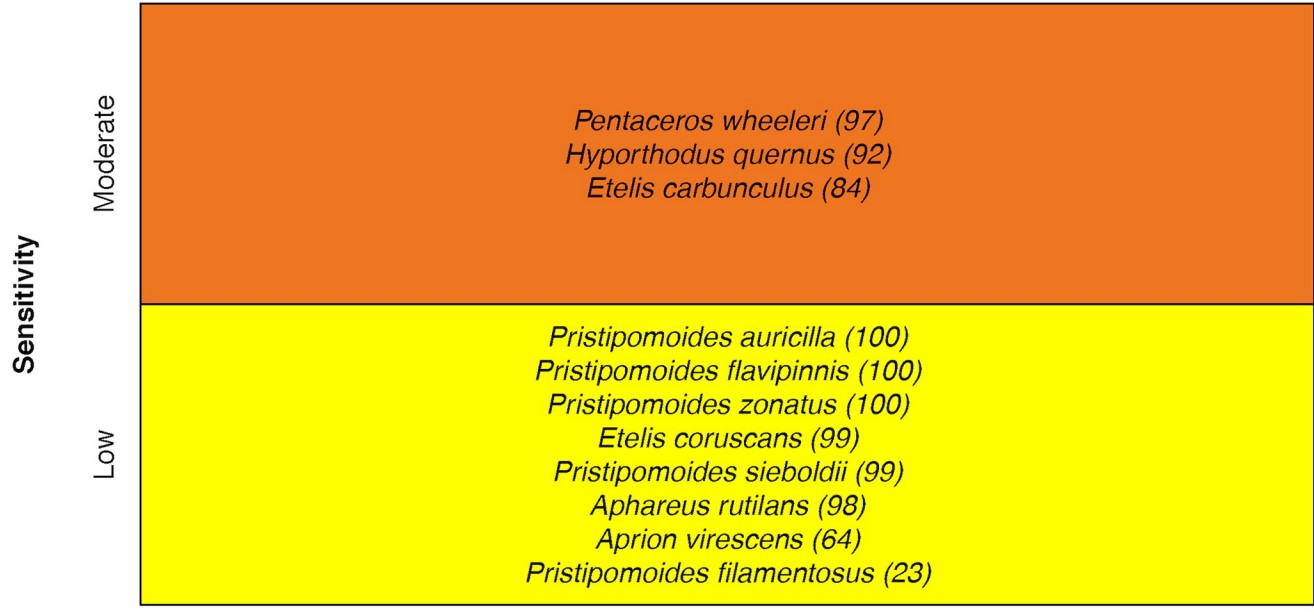

**Fig 12. Vulnerability scores of deep-slope species.** Overall climate vulnerability, as determined by point estimates, is denoted by box background color: Moderate (yellow), and High (orange). Certainty in vulnerability score is denoted by parenthetical number following the species name, indicating bootstrapped scores (0–100% certainty).

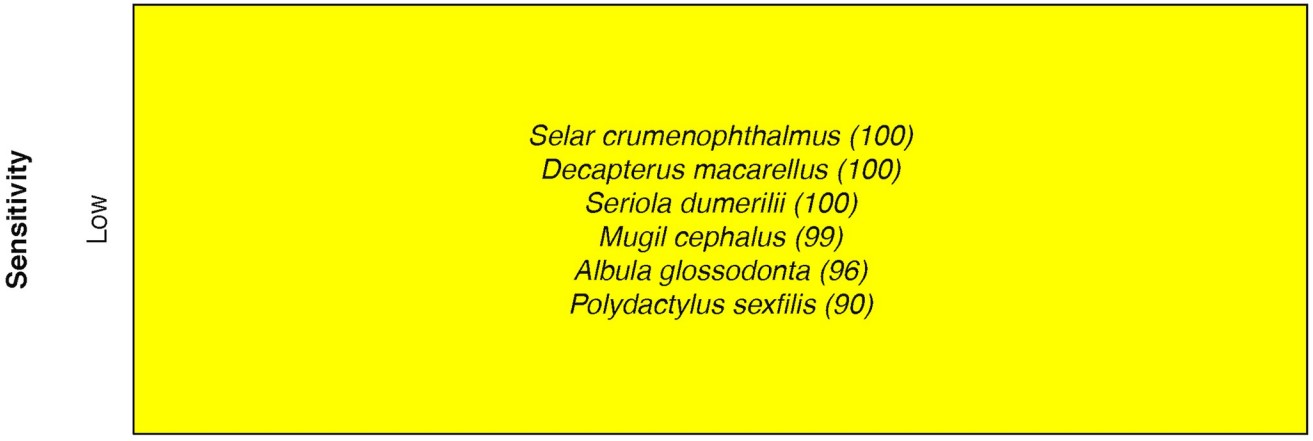

**Fig 13. Vulnerability scores of coastal species.** Overall climate vulnerability, as determined by point estimates, is denoted by box background color: Moderate (yellow). Certainty in vulnerability score is denoted by parenthetical number following the species name, indicating bootstrapped scores (0–100% certainty).

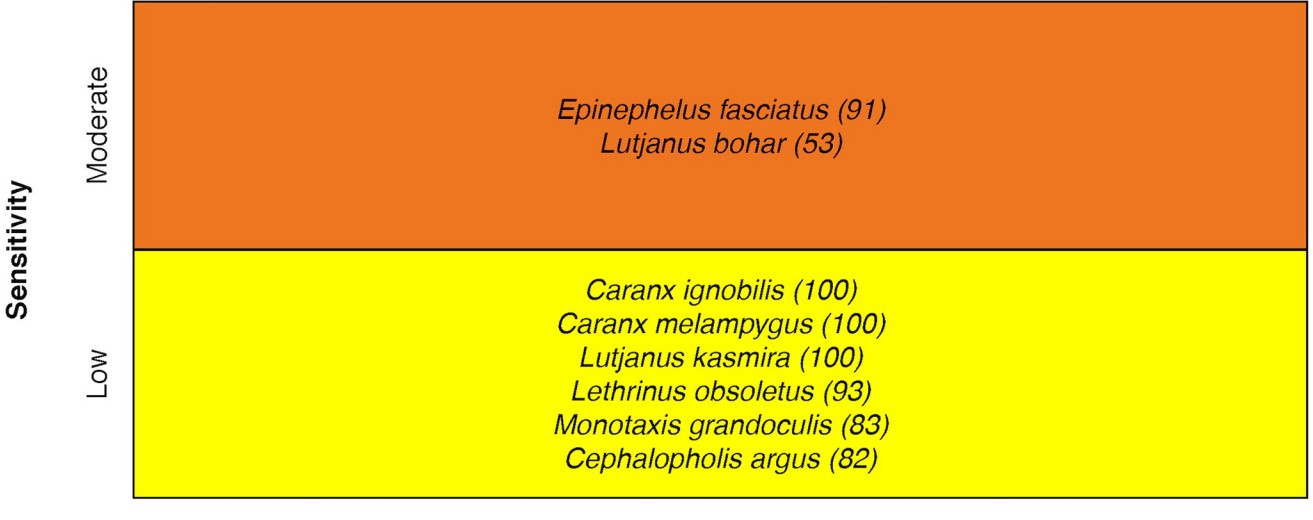

**Fig 14. Vulnerability scores of Jack, Emperor, Grouper, and Snapper species.** Overall climate vulnerability, as determined by point estimates, is denoted by box background color: Moderate (yellow), and High (orange). Certainty in vulnerability score is denoted by parenthetical number following the species name, indicating bootstrapped scores (0–100% certainty).

could also be a habitat threat for pelagic species ([52] but see [53]). However, some pelagic species are relatively tolerant of low oxygen concentrations [54, 55]. For example, among the tunas, Bigeye Tuna are the most tolerant of low oxygen concentrations and can dive to depths with oxygen as low as 1–1.5 mL $O_2$/L while Skipjack and Yellowfin generally don't spend time in waters with levels below 3.5mL $O_2$/L [54, 55]. Oxygen tolerance and environmental habitat covariates for Bigeye Tuna are relatively well studied by a combination of laboratory studies [54, 55], field studies [56, 57], and comprehensive habitat modeling efforts [58]. This example suite of studies reflects the requisite multifaceted approach needed for better understanding climate change impacts to pelagic species.

**Coral reef fishes.** The sensitivity scores of many coral reef fishes ranged between Low and Moderate. This is intuitively surprising, given that many coral reef species depend on a biogenic habitat that is itself extremely threatened by climate change [59–62]. There are also potential behavioral and physiological effects of temperature and ocean acidification on coral reef fishes, which could include increased boldness, decreased fecundity, decreased offspring quality, and impaired sensory abilities [63–66]. Sensory abilities are particularly important for navigation and homing during early life stages when the larvae settle from the plankton as new recruits to adult habitat [67].

In particular, the Bumphead Parrotfish has very specific recruitment requirements and so emerges as a more sensitive species [68]. They recruit to acroporid corals in shallow lagoon areas, habitats that are likely to be severely affected by climate change [61, 69].

Several factors contributed to the deceptively low sensitivity scores for coral reef fishes: (1) the relatively short (37-year) timespan considered, (2) the decision that life history complexity could increase resiliency instead of decreasing it, (3) consideration that some coral species are more resilient to ocean acidification than often assumed [70], and (4) the assignment of a moderate score to sensitivity attributes when information was insufficient to give informed

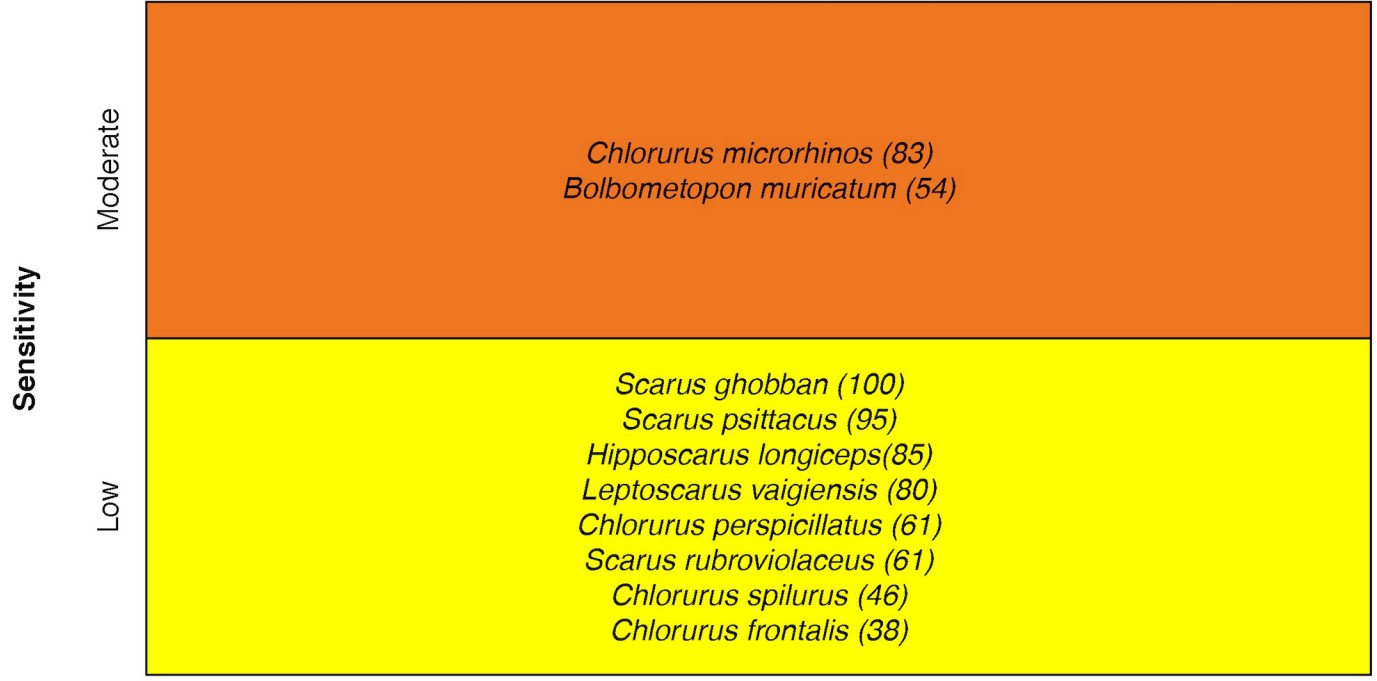

**Fig 15. Vulnerability scores for parrotfish.** Overall climate vulnerability, as determined by point estimates, is denoted by box background color: Moderate (yellow) and High (orange). Certainty in vulnerability score is denoted by parenthetical number following the species name, indicating bootstrapped scores (0–100% certainty).

rankings. We emphasize that with a more precautionary approach to scoring when quantitative information was not available, the overall coral reef fishes sensitivity scores would be considerably higher.

**Invertebrate species.** Invertebrates face many unique challenges from climate change. In particular, given their small home range sizes and limited adult mobility, they are highly susceptible to regional impacts.

In Hawai'i, the Hawaiian yellow-foot limpet was ranked the highest in vulnerability. Not only is it endemic to the Hawaiian Archipelago, it also has a particular habitat requirement of crustose coralline algae (CCA) on intertidal basalt rock in areas that experience high wave action. In addition, Hawaiian yellow-foot limpets rely on a heavily harvested co-occurring grazer to create habitable space [71–73].

More generally, the relatively less-mobile invertebrates are more likely to experience Allee effects at low population numbers, which could negate or slow recovery [74, 75]. The Allee effect describes a decline in individual fitness at low population densities. In the case of broadcast spawning invertebrates, poor fertilization occurs at very low densities. Below a critical density threshold, populations can crash to extinction [74]. Although fishery-depleted populations of the surf redfish *Actinopyga mauritiana* on Saipan showed recovery in a survey conducted nine years post-harvest [76], Friedman et al. [77] noted that many sea cucumber populations had not recovered decades following over-harvest.

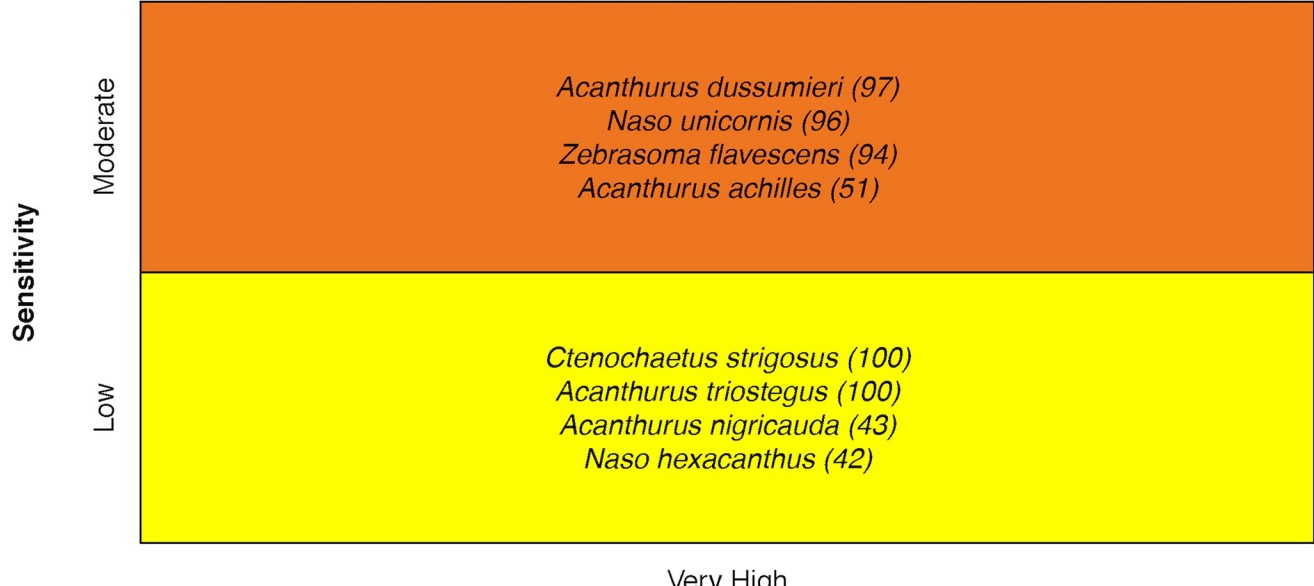

**Fig 16. Vulnerability scores of coral reef surgeonfishes.** Overall climate vulnerability, as determined by point estimates, is denoted by box background color: Moderate (yellow), and High (orange). Certainty in vulnerability score is denoted by parenthetical number following the species name, indicating bootstrapped scores (0–100% certainty).

However, some invertebrates may benefit from changes in their habitat. For example, the Samoan crab *Scylla serrata* might thrive in disturbed habitats. This species was identified as a potential "winner" in climate change scenarios given its quick acclimation to Hawaiʻi after introduction [78], and its low sensitivity rankings. This species is relatively mobile compared to other invertebrates and is not specialized in most aspects of its life-history.

**Methodology considerations.**   The Pacific Islands Vulnerability Assessment (PIVA) is complex, given the range of taxa involved in the study, and the diversity and extent of coral reef habitats. Reef fishes are highly susceptible to ocean acidification, both directly in terms of physiological and behavioral effects [61, 64, 66], and indirectly in terms of habitat and prey loss [59, 79]. This assessment scored many reef fish as Low to Moderate in vulnerability, which is due to key considerations in the methodology that set the PIVA apart from the original vulnerability assessment developed in the Northeast US.

First, in the Northeast US, many of the species considered are managed stocks, with detailed assessments of stock status and available data on life history parameters. In the Pacific Islands region, many coral reef and invertebrate species are not quantitatively assessed and lack detailed and spatially appropriate life history information.

The low vulnerability scores for species in the Pacific region reflect uncertainty in sensitivity attributes rather than actual biological coping mechanisms of the species. For example, Other Stressors (including habitat degradation, pollution, disease, or changes in the food web) was ranked as highly important but with low data quality for many coral reef species. Likewise, Early Life History Survival and Settlement Requirements was ranked as extremely important to assess climate impacts, yet data-quality was consistently ranked as low for Pacific Island region species. The default assignment of a Moderate score to sensitivity attributes of individual species in the instances when information was insufficient to make informed assignments

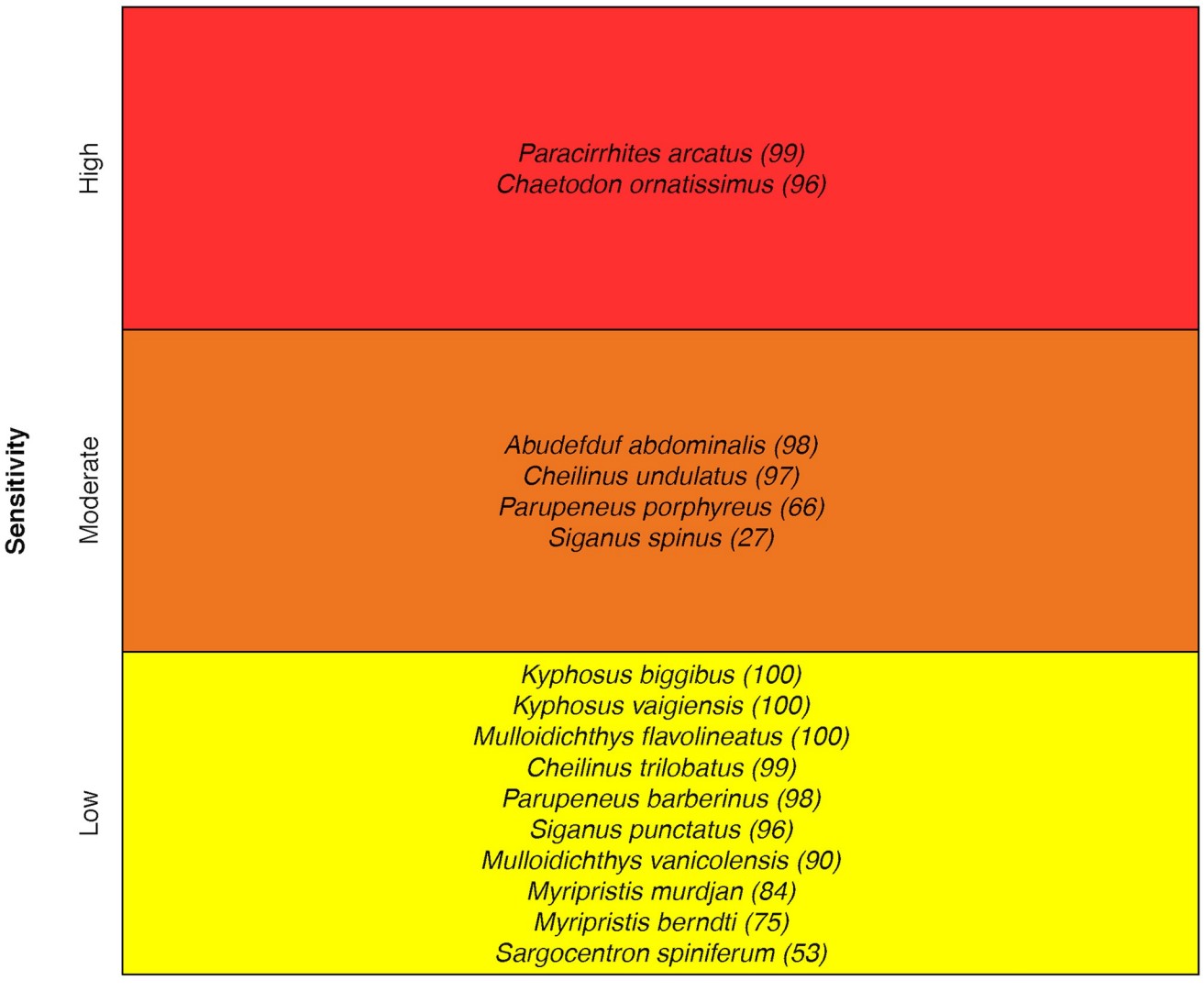

**Fig 17. Vulnerability scores for 'other' coral reef fish species.** Overall climate vulnerability, as determined by point estimates, is denoted by box background color: Moderate (yellow), High (orange), and Very High (red). Certainty in vulnerability score is denoted by parenthetical number following the species name, indicating bootstrapped scores (0–100% certainty).

may have driven the vulnerabilities toward Moderate ranks for many species. Habitat assessments (see [80]) can complement PIVA and in particular improve our assessment of coral reef species. Additionally, PIVA would be greatly supplemented by an integrated assessment of habitat, species, ecological communities, and social communities (e.g., recreational fishers, subsistence fishers, community-based managers) [80, 81].

Second, some coral reef species, such as surgeonfishes, do not fit well into standard categories, making it difficult to interpret sensitivity from life history attributes. For example, the reproductive schedule of many surgeonfishes trades output for long life, with mass

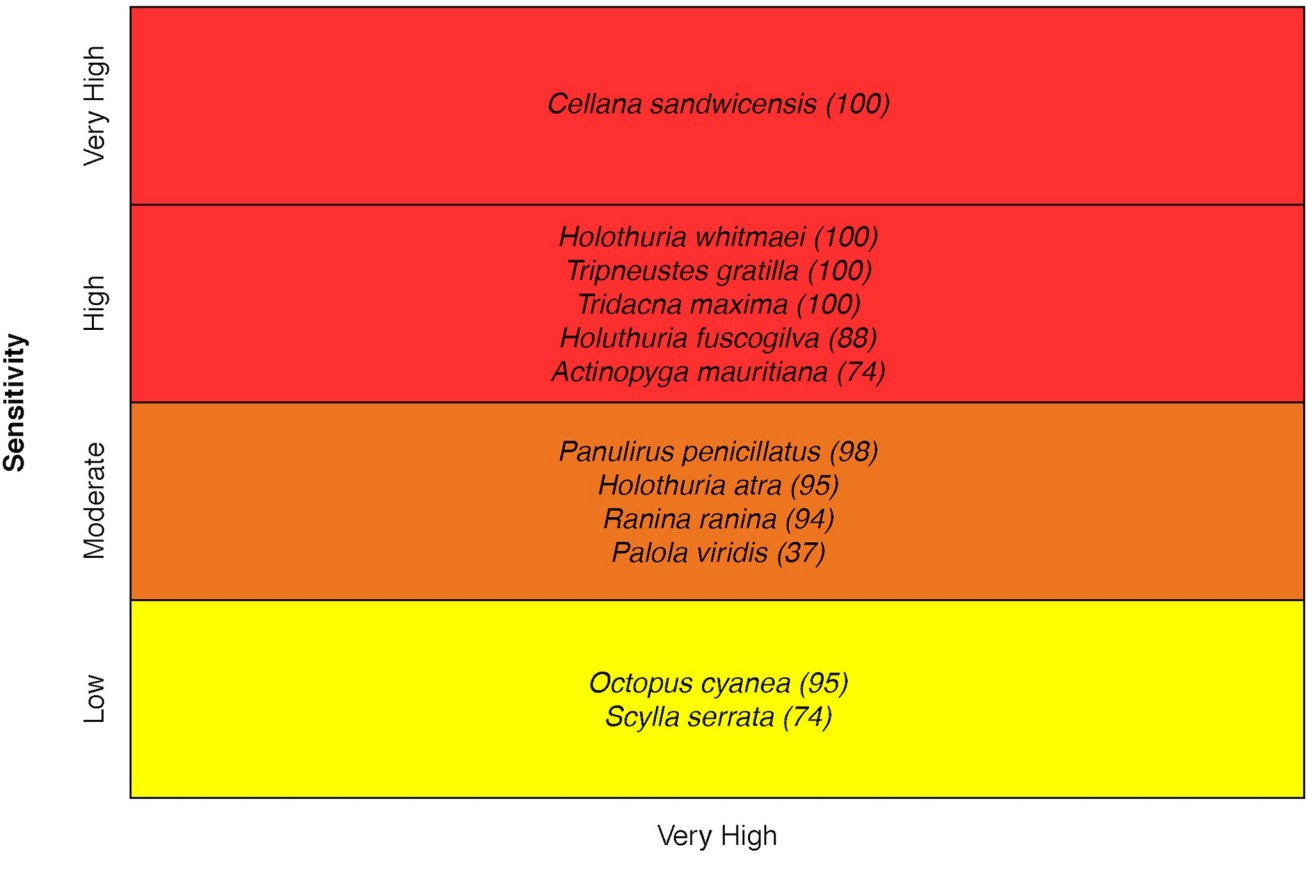

**Fig 18. Vulnerability scores for invertebrate species.** Overall climate vulnerability, as determined by point estimates, is denoted by box background color: Moderate (yellow), High (orange), and Very High (red). Certainty in vulnerability score is denoted by parenthetical number following the species name, indicating bootstrapped scores (0–100% certainty).

recruitments at sporadic times, making it difficult to assess population growth rate in a standard way [82].

Third, coral reef fishes have added complexity in reproductive strategy in that some species are sequential hermaphrodites [83]. The expression of one sexor the other is partly dependent on habitat quality as well as population density [84, 85]. Because experts for this PIVA generally regarded hermaphroditism, a characteristic that makes reproduction more complex, as potentially beneficial with respect to a changing environment, future RVAs should perhaps redefine Complexity in Reproductive Strategy to Flexibility in Reproductive Strategy. In contrast, Hare et al. [16] interpreted high complexity as solely disadvantageous. In this PIVA, Complexity in Reproductive Strategy was potentially bi-directional which should be considered when interpreting scoring for this attribute.

Fourth, the biological sensitivity attributes were developed for finfishes, making the invertebrate species particularly difficult to assess. The life history for invertebrates can change dramatically among locations. For example, for the sea cucumbers, asexual reproduction occurs in some, but not all locations [86]. Likewise, the spawning cycle can be different across locations [87].

Fifth, in scoring, considering short-term acclimation or adaptive capacity of many coral reef fishes gravitated the sensitivity scores to Low and Moderate. However, this consideration of short-term acclimation or adaptive capacity would have more bearing if the time frame considered for the assessment was expanded beyond the year 2055.

Overall, this PIVA is a relative analysis for a subset of species across the Pacific Island region and does not experimentally evaluate the direct effect of climate change on these species. It should be used with these caveats in mind and should definitely not be interpreted to conclude that ocean warming and acidification will have few effects on reef fish species in the Pacific.

**Exposure variables.**   The variety of exposure variables included in this type of analysis is ultimately constrained by what datasets are available over climate-change time horizons. Furthermore, many variables are spatially represented at scales inappropriate for the target species. For example, we included an intertidal limpet in PIVA yet applied many of the same exposure variables to it as a pelagic tuna, albeit summarized over the habitat pixels respectively involved. While this type of potential mismatch is unfortunate, we feel that the PIVA results are a worthwhile first step towards understanding the impacts of climate change on a wide range of marine species occupying a wide range of habitats across the Pacific, and that subsequent targeted research can build upon the PIVA findings.

Exposure variables were scored under the assumption that perturbations to baseline conditions are equally deleterious whether the departures were negative or positive. For example, high oxygen concentrations were scored similarly to low oxygen concentrations. Until the specific dose-response functions are fully mapped out for each species, these types of assumptions will remain in the scoring process but should be carefully considered. Fortunately, situations such as this, e.g., where high oxygen values had a substantial habitat footprint, did not manifest in our results nor drive any taxa's overall vulnerability to climate change projections.

As mentioned previously, the final exposure scores were influenced both by predicted changes in magnitude of a variable as well as scaling by historical variability in that variable. Some exposure variables, such as current velocity and precipitation, exhibit substantial historical variability that tends to dampen the exposure score for these variables even if quite high changes in magnitude over time are anticipated (Fig 2). Some exposure variables such as surface pH, surface oxygen concentration, salinity (both surface and bottom), and surface temperature have relatively stationary spatial signals across the Pacific with relatively little de-trended variability which, being in the denominator of the calculation, tends to inflate the overall exposure score irrespective of predicted change in magnitude over time. Small changes in magnitude are therefore considered more impactful with variables that exhibit such spatial and temporal uniformity in the background environment. Other exposure variables such as mixed layer depth, bottom temperature, primary productivity, and chlorophyll exhibit the expected relationship of higher exposure scores with higher negative or positive perturbations relative to historical values. This contrast in how different variables impact the exposure scoring approach used here highlights the importance of an exact taxon-specific matching of geographic distributions and exposure fields, which this vulnerability exercise accomplished as an improvement over the established protocol which relied on visual scrutiny of exposure variable maps and species range maps. Geographic distributions in this exercise ranged from archipelagic endemics to circumglobal, which made it difficult for experts to glean quantitative scoring metrics from simple exposure variable maps. The automated subsetting and tabulation of exposure fields therefore allowed an objective and replicable exposure scoring to take place in this RVA. This is a key improvement to the original methodology posed for RVAs and this approach for incorporating exposure variables should be applied to future RVAs as well.

## Data needs

Many Pacific species were difficult to assess given unknowns in their biology and ecology. In particular for finfishes, the early life history and larval stage proved the most challenging to assess given unknowns about their requirements. For example, increasing evidence shows that ocean acidification, besides being deleterious for corals, negatively affects the sensory capabilities and behavior of larval coral-reef fishes [64, 88–92]. Recent investigation of microplastic pollution ingestion suggests sublethal effects during early life stages [93, 94]. The temporal and spatial overlaps between critical early life history stages and plastics remain unknown at this time but of clear importance to monitor particularly in the neuston and other environmentally dynamic epipelagic nursery habitats. With more species-specific information on this topic, it seems likely this group of fishes would have been scored as more vulnerable to climate change.

Some species in the invertebrate group were particularly challenging to assess, given the uncertainty in basic knowledge about their life history and distribution. Finally, the role of adaptation and the ability of species to increase fitness apace with climate change was a question that emerged that we need to address to better understand the vulnerability of marine life to climate change.

Even with these non-trivial data needs, the Rapid Vulnerability Assessment framework has value as a method to synthesize what is known and transform disparate information into a usable format to guide management and further research. Climate change is already affecting marine species and ecosystems [95, 96]. While there is much to be discovered about species sensitivities and exposures, there is also little time to implement climate-informed ecosystem management. This study addressed the questions of which species, among 83 examples, are most vulnerable to climate change and where science and management should focus efforts to reduce these risks in the Pacific Islands region. We provided a relative climate vulnerability ranking across species, identified key attributes and factors that drive this vulnerability, and identified key data gaps to guide future research.

## Supporting information

**S1 Table. List of species in each functional group.**
(TIFF)

**S2 Table. List of species in subcategories of coral reef functional group.**
(TIFF)

**S1 File. Description of exposure scores and code.**
(DOCX)

**S2 File. Overall vulnerability scores for all species.**
(PDF)

**S3 File. Bootstrapped sensitivity scores per species.**
(PDF)

**S1 Fig. Overall vulnerability rank as determined by the product of exposure and sensitivity component scores.** The possible range of these scores is between 1 and 16. The numerical values for the climate vulnerability rank are as follows: 1–3 Low (green), 4–6 Moderate (yellow), 8–9 High (orange), and 12–16 Very High (red).
(TIF)

## Acknowledgments

We thank Roger Griffis, Michael Seki, Evan Howell, Frank Parrish, Rusty Brainard, Jeff Hare, Melanie Abecassis, Brittany Huntington, and Tye Kindinger for supporting this work, and thank Howard Choat and Ivor Williams for their valuable contributions to the assessment. We thank Matthew Iacchei, Frank Parrish, and Ryan Rykaczewski for their reviews of the manuscript. This is the Ocean Research Explorations Hawaiian Islands Biodiversity Project publication 07.

## Author Contributions

**Conceptualization:** Donald R. Kobayashi.

**Formal analysis:** Donald R. Kobayashi, Mark Nelson.

**Investigation:** Jonatha Giddens, Donald R. Kobayashi, Jacob Asher, Charles Birkeland, Mark Fitchett, Mark A. Hixon, Melanie Hutchinson, Bruce C. Mundy, Joseph M. O'Malley, Marlowe Sabater, Molly Scott, Jennifer Stahl, Rob Toonen, Michael Trianni, Phoebe A. Woodworth-Jefcoats, Johanna L. K. Wren.

**Methodology:** Mark Nelson.

**Project administration:** Mark Nelson.

**Supervision:** Donald R. Kobayashi.

**Visualization:** Jonatha Giddens, Gabriella N. M. Mukai.

**Writing – original draft:** Jonatha Giddens, Donald R. Kobayashi.

**Writing – review & editing:** Donald R. Kobayashi, Gabriella N. M. Mukai.

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
