## [Decision Letter · Decision Letter 0]

25 Apr 2022

PONE-D-22-05660Assessing the vulnerability of marine life to climate-change in the Pacific Islands RegionPLOS ONE

Dear Dr. Kobayashi,

Thank you for submitting your manuscript to PLOS ONE. After careful consideration, we feel that it has merit but does not fully meet PLOS ONE’s publication criteria as it currently stands. Therefore, we invite you to submit a revised version of the manuscript that addresses the points raised during the review process.

Overall, this manuscript is well written and acceptable for publication after minor revisions.  The methods follow previous peer-reviewed protocols, the complex results are synthesized in detail, and logical high-level conclusions are presented.  In addition to the comments and editorial suggestions from the reviewers, please address these four improvements to the methods.

Highlight differences between the study region and other previously assessed regions in the U.S.: Although this ms mostly follows the protocol defined by efforts from previously completed regions, the authors highlight some challenges specific to this region, which is quite different from other U.S. marine ecosystems.  These methodological differences could be highlighted in the text.Explain the methods more clearly: The methods section can be hard to understand at times, so I urge the authors to address the reviewer comments to address these limitations. In addition to describing some of the methods in more detail, I wonder if you could walk the readers through some specific examples for some taxa that either scored especially high / low, or that proved particularly challenging to assess.Define exposure scores: It is difficult to determine how the final exposure score for a given attribute, was determined. Based on the text and in the supplement, it seems that it is based on a percentage of grid cells that have different exposure levels.  However, it is important to explicitly describe what the different levels mean.  For instance, how are the “low” and “medium” levels mean and how are they defined quantitively:  how do they relate to the baseline.  Please consider adding a table where each of these levels is clearly defined.Define the expert scoring system: For the “sensitivity” section, the framework of the expert scoring is unclear? Can you please explain what are the range of possible values that can be assigned for a given factor?  Can you also provide an explicit description of what the various individual values mean or represent?

We look forward to receiving your revised manuscript.

Kind regards,

David Hyrenbach, Ph.D.

Academic Editor

PLOS ONE

Journal Requirements:

2.We note that you have stated that you will provide repository information for your data at acceptance. Should your manuscript be accepted for publication, we will hold it until you provide the relevant accession numbers or DOIs necessary to access your data. If you wish to make changes to your Data Availability statement, please describe these changes in your cover letter and we will update your Data Availability statement to reflect the information you provide.

3. We note that in Figure 2, 5, 7, 8, 9, 10 and Supporting Information 1 Exposure Code in your submission contain map images which may be copyrighted. All PLOS content is published under the Creative Commons Attribution License (CC BY 4.0), which means that the manuscript, images, and Supporting Information files will be freely available online, and any third party is permitted to access, download, copy, distribute, and use these materials in any way, even commercially, with proper attribution. For these reasons, we cannot publish previously copyrighted maps or satellite images created using proprietary data, such as Google software (Google Maps, Street View, and Earth). For more information, see our copyright guidelines: http://journals.plos.org/plosone/s/licenses-and-copyright.

a. You may seek permission from the original copyright holder of Figure 2, 5, 7, 8, 9, 10 and Supporting Information 1 Exposure Code to publish the content specifically under the CC BY 4.0 license.  

(We thank Roger Griffis, Michael Seki, Evan Howell, Frank Parrish, Rusty Brainard, Jeff Hare, Melanie Abecassis, Brittany Huntington, and Tye Kindinger for supporting this work, and thank Howard Choat and Ivor Williams for their valuable contributions to the assessment. We thank Matthew Iacchei, Frank Parrish, and Ryan Rykaczewski for their reviews of the manuscript. Funding for this work was provided by the NOAA Office of Science and Technology, NOAA Pacific Islands Fisheries Science Center, and the Joint Institute of Marine and Atmospheric Research at the University of Hawai‘i, Mānoa. Travel for Howard Choat to the Expert Panel Workshop was provided by the Joint Institute of Marine and Atmospheric Research at the University of Hawai‘i, Mānoa Visiting Science Award. This is the Ocean Research Explorations Hawaiian Islands Biodiversity Project publication 07.)

(The author(s) received no specific funding for this work.)

Reviewers' comments:

Reviewer's Responses to Questions

**Comments to the Author**

1. Is the manuscript technically sound, and do the data support the conclusions?

Reviewer #1: Yes

Reviewer #2: Yes

2. Has the statistical analysis been performed appropriately and rigorously? 

Reviewer #1: Yes

Reviewer #2: Yes

3. Have the authors made all data underlying the findings in their manuscript fully available?

Reviewer #1: Yes

Reviewer #2: Yes

4. Is the manuscript presented in an intelligible fashion and written in standard English?

Reviewer #1: Yes

Reviewer #2: Yes

5. Review Comments to the Author

Reviewer #1: The authors present a climate vulnerability assessment for 83 marine fish and invertebrate species in the Pacific Islands. They use a previously published approach and provide a good balance of describing the approach in enough detail for understanding without repeating all the details from prior papers. They also do a good job explaining changes to the approach necessary for their application. The authors find high exposure to climate change across all species assessed and find a range of sensitivities across species. They describe their results from a functional group perspective. The results are important because they identify species with relatively higher vulnerability to climate change, they identify factors of climate change important in the Pacific Islands, and they identify important data gaps.

Overall, the manuscript is well done and acceptable after minor revision. The manuscript is well written and the information well presented. The methods follow previous peer-reviewed procedures. The complex results are synthesized and high-level conclusions presented. My comments are minor.

Line 281 - is a -1.37% decrease is pH meaningful given that the pH scale is logarithmic?

Line 450 - I recommend moving this summary paragraph before the functional groups (at line

356)

Line 462 - The authors describe the high uncertainty for 8 species. A sentence or two on the magnitude of low uncertainty would be useful or a sentence summarizing the number of species with high, moderate, and low uncertainty.

Line 598 - Worth mentioning that the analysis is relative.

Given the importance of habitat, it could be useful for the authors to briefly discuss Farr et al (2021) and the potential value of assessing climate change impacts on habitats. https://journals.plos.org/plosone/article?id=10.1371/journal.pone.0260654

Figure 1 could be included in a table.

Figure 3 could be in supplemental materials

Reviewer #2: In the manuscript by Giddens et al., 80+ species from the U.S. jurisdiction islands in the Pacific Ocean are assessed qualitatively for their sensitivity to climate change, and their potential exposure to climate variables was projected to the mid 21st century. This analysis is part of the effort by NOAA to rapidly assess marine species in all regions of its governance for vulnerability to climate change. As such, it is mostly following the protocol defined by efforts in the previously completed regions. Although the authors highlight some challenges associated with this format, that are specific to this region, which is quite different from other U.S. managed marine ecosystems. This manuscript was well written and I think it is quite close to being ready for publication. I commend the authors for putting together a polished submission. Please see comments below.

GENERAL COMMENT: The methods section can be a little frustrating to understand at times. Some added text to better describe some of the methods would make a big difference. Specific areas in the methods are highlighted below with line comments.

L85-87: This sentence reads somewhat awkwardly. Recommend reformatting, perhaps starting with something like “The Pacific region offered unique challenges including…….”

L142-143: Here and elsewhere in the manuscript, websites should be more formally described (or at least give the title of the project/database) and also cited. Often in the methods, the sources are only given as a website in parentheses.

L146-148: A little more detail (1-2 sentences) should be added for the climate projection data. Provide some context to “model runs” and give a brief description of what the RCP 8.5 scenario represents.

L151-152: Is the historical reference period also from the same climate models? In other words, not from an independent data set. If so, probably worth noting that.

L187-192: It is still a bit difficult to determine how the final exposure score, for a given attribute, is determined. Based on text here and in the supplement, it seems that it is based on a percentage of grid cells that are low, medium, etc. exposure. I also assume that “low” means within one standard deviation of the baseline, while “medium” is two standard deviations, etc. However, this doesn’t appear to be described anywhere.

L203: For the “Sensitivity” section, not sure if I missed it (sorry if so), but is it stated anywhere what the framework of the expert scoring is? What are the range of possible values that can be assigned for a given factor?

Table-1 heading is missing “Northern”

L212: Change “their” to “them”

L587-588: Was life history complexity treated differently in the other regions where this was done?

6. PLOS authors have the option to publish the peer review history of their article (what does this mean?). If published, this will include your full peer review and any attached files.

Reviewer #1: No

Reviewer #2: No

---

## [Author Response · Author response to Decision Letter 0]

2 Jun 2022

Please see attachment, but here is a copy/paste of the relevant material.

Please find below our bulleted responses to the review of our manuscript. The Editor comments are first, followed by Reviewer 1, Reviewer 2. We have addressed every item that was brought to our attention.

Thx. Donald Kobayashi, Gabriella Mukai & the PIVA team

Editor

1. Highlight differences between the study region and other previously assessed regions in the U.S.: Although this ms mostly follows the protocol defined by efforts from previously completed regions, the authors highlight some challenges specific to this region, which is quite different from other U.S. marine ecosystems. These methodological differences could be highlighted in the text.

○ Changes were made in the introduction and discussion to highlight these challenges and solutions (e.g., quantifying exposure, discussion of hermaphroditism). 

2. Explain the methods more clearly: The methods section can be hard to understand at times, so I urge the authors to address the reviewer comments to address these limitations. In addition to describing some of the methods in more detail, I wonder if you could walk the readers through some specific examples for some taxa that either scored especially high / low, or that proved particularly challenging to assess.

○ Exposure and sensitivity scoring methodologies were clarified in their respective sections and in the S1 Materials, including enough detail to enable readers to understand vulnerability scoring. 

3. Define exposure scores: It is difficult to determine how the final exposure score for a given attribute, was determined. Based on the text and in the supplement, it seems that it is based on a percentage of grid cells that have different exposure levels. However, it is important to explicitly describe what the different levels mean. For instance, how are the “low” and “medium” levels mean and how are they defined quantitively: how do they relate to the baseline. Please consider adding a table where each of these levels is clearly defined.

○ Exposure scoring was clarified in the main text and we also provided more details in S1 Materials. A table within S1 Materials provides the cutoff values for each score. 

4. Define the expert scoring system: For the “sensitivity” section, the framework of the expert scoring is unclear? Can you please explain what are the range of possible values that can be assigned for a given factor? Can you also provide an explicit description of what the various individual values mean or represent?

○ Sensitivity scoring was clarified in the main text. Score values were also provided in the text.

Changes to the reference list include: citations for the web portals (#22-28), the NOAA data reports for each species group (#30-35), removal of one citation (#38) since this was incorporated into citations 30-35, an addition of one citation (#80) per the suggestions of a reviewer and another citation (#81) that complements the citation 80. 

Maps were generated by the authors using Generic Mapping Tools (GMT, freeware mapping software) and should not fall under copyright restrictions. 

Reviewer 1

1. Line 281 - is a -1.37% decrease is pH meaningful given that the pH scale is logarithmic?

○ Originally, we had standardized all the changes/anomalies to percentage. Reviewer 1 brings up an excellent point that describing the change in pH would be more meaningful and clearer if the change was described in pH units rather than percentage. The text has been modified to reflect the changes in pH units and surface oxygen units. 

2. Line 450 - I recommend moving this summary paragraph before the functional groups (at line 356)

○ This suggestion was incorporated. The paragraph was moved to the suggested place along with the figure. Following figures were relabeled. 

3. Line 462 - The authors describe the high uncertainty for 8 species. A sentence or two on the magnitude of low uncertainty would be useful or a sentence summarizing the number of species with high, moderate, and low uncertainty.

○ We added a sentence on the breakdown of high and moderate certainty and updated the low certainty species list. We also changed the wording to say “certainty” (instead of “uncertainty”) since we think it’s more intuitive to associate a high number value (i.e., 90%) with high certainty rather than low uncertainty. 

4. Line 598 - Worth mentioning that the analysis is relative. 

○ This suggestion was incorporated and we further clarified the limitations of this analysis. 

5. Given the importance of habitat, it could be useful for the authors to briefly discuss Farr et al (2021) and the potential value of assessing climate change impacts on habitats. https://journals.plos.org/plosone/article?id=10.1371/journal.pone.0260654

○ This suggestion was incorporated in addition to a brief discussion of assessing the impacts on human communities. 

6. Figure 1 could be included in a table.

○ This suggestion was incorporated. Figure 1 was converted to a table (Table 1). Following tables and figures were relabeled as appropriate. 

7. Figure 3 could be in supplemental materials. 

○ This suggestion was incorporated. Figure 3 was moved to supplementals. Following figures were relabeled as appropriate. 

Reviewer 2 

1. L85-87: This sentence reads somewhat awkwardly. Recommend reformatting, perhaps starting with something like “The Pacific region offered unique challenges including…….”

○ This suggestion was incorporated.

2. L142-143: Here and elsewhere in the manuscript, websites should be more formally described (or at least give the title of the project/database) and also cited. Often in the methods, the sources are only given as a website in parentheses.

○ This suggestion was incorporated. 

3. L146-148: A little more detail (1-2 sentences) should be added for the climate projection data. Provide some context to “model runs” and give a brief description of what the RCP 8.5 scenario represents.

○ This suggestion was incorporated. RCP8.5 and “model runs” were defined. 

4. L151-152: Is the historical reference period also from the same climate models? In other words, not from an independent data set. If so, probably worth noting that.

○ Yes, and we explicitly stated this in the text now. 

5. L187-192: It is still a bit difficult to determine how the final exposure score, for a given attribute, is determined. Based on text here and in the supplement, it seems that it is based on a percentage of grid cells that are low, medium, etc. exposure. I also assume that “low” means within one standard deviation of the baseline, while “medium” is two standard deviations, etc. However, this doesn’t appear to be described anywhere.

○ Thank you for your comment, this suggestion was incorporated in the main text and we also provided more details in S1 Materials. 

6. L203: For the “Sensitivity” section, not sure if I missed it (sorry if so), but is it stated anywhere what the framework of the expert scoring is? What are the range of possible values that can be assigned for a given factor?

○ Thank you for your comment, the text has been revised to better explain how experts scored each attribute and the values of each score.

7. Table-1 heading is missing “Northern”

○ This suggestion was incorporated.

8. L212: Change “their” to “them”

○ This suggestion was incorporated.

9. L587-588: Was life history complexity treated differently in the other regions where this was done?

○ Thank you for your comment, we expanded on the challenge of scoring hermaphroditic species and working within the confines of the “Complexity of Reproductive Strategy” attribute. This challenge was not addressed in the original Northeast region RVA.

---

## [Editor Report · Decision Letter 1]

21 Jun 2022

Assessing the vulnerability of marine life to climate change in the Pacific Islands region

PONE-D-22-05660R1

Dear Dr. Kobayashi,

We’re pleased to inform you that your manuscript has been judged scientifically suitable for publication and will be formally accepted for publication once it meets all outstanding technical requirements.

Upon reviewing the ms, I would like to point out two minor typographical errors in the Literature Cited section:

1)  Line 840 is left blank and #49  in line 841 is highlighted in yellow

2) In Line 870:  Reference #59 is crossed out and highlighted in red

I would also like to thank you the detailed description of the data and software you used to generate your maps.  

Pending the green light from the editorial office, I would suggest you add this detailed explanation (and the supporting references) in the supplementary materials.   

Otherwise, I thank you for addressing all the reviewer comments so thoroughly.

Kind regards,

David Hyrenbach, Ph.D.

Academic Editor

PLOS ONE
---

## [Editor Report · Acceptance letter]

29 Jun 2022

PONE-D-22-05660R1 

Assessing the vulnerability of marine life to climate change in the Pacific Islands region 

Dear Dr. Kobayashi:

I'm pleased to inform you that your manuscript has been deemed suitable for publication in PLOS ONE. Congratulations! Your manuscript is now with our production department. 

Kind regards, 

on behalf of

Dr. David Hyrenbach 

Academic Editor

PLOS ONE